# Dipolar extracellular potentials generated by axonal projections

**Thomas McColgan[1]\*, Ji Liu[2], Paula Tuulia Kuokkanen[1,3], Catherine Emily Carr[2], Hermann Wagner[4], Richard Kempter[1,3,5]\***

[1]Department for Biology, Institute for Theoretical Biology, Humboldt-Universität zu Berlin, Berlin, Germany; [2]Department of Biology, University of Maryland, College Park, United States; [3]Bernstein Center for Computational Neuroscience, Berlin, Germany; [4]Institute for Biology II, RWTH Aachen, Aachen, Germany; [5]Einstein Center for Neurosciences, Berlin, Germany

**Abstract** Extracellular field potentials (EFPs) are an important source of information in neuroscience, but their physiological basis is in many cases still a matter of debate. Axonal sources are typically discounted in modeling and data analysis because their contributions are assumed to be negligible. Here, we established experimentally and theoretically that contributions of axons to EFPs can be significant. Modeling action potentials propagating along axons, we showed that EFPs were prominent in the presence of terminal zones where axons branch and terminate in close succession, as found in many brain regions. Our models predicted a dipolar far field and a polarity reversal at the center of the terminal zone. We confirmed these predictions using EFPs from the barn owl auditory brainstem where we recorded in nucleus laminaris using a multielectrode array. These results demonstrate that axonal terminal zones can produce EFPs with considerable amplitude and spatial reach.

DOI: https://doi.org/10.7554/eLife.26106.001

**\*For correspondence:**
thomas.mccolgan@gmail.com
(TMC);
r.kempter@biologie.hu-berlin.de
(RK)

## Introduction

Extracellular field potentials (EFPs) are at the heart of many experimental approaches used to examine the inner workings of the brain. Types of EFPs include invasively recorded signals such as the electrocorticogram (ECoG) and the local field potential (LFP), as well as the noninvasively recorded electroencephalogram (EEG) and the auditory brainstem response (ABR) (*Brette and Destexhe, 2012*; *Nunez and Srinivasan, 2006*). Measures derived from the EFP such as the current source density (CSD) and multiunit activity (MUA) are also frequently used. The origins of these signals and measures, especially in cases in which the activity is not clearly attributable to a single cell, is a matter of debate (*Buzsáki et al., 2012*).

EFPs in the brain were long thought to be primarily of synaptic origin (*Buzsáki et al., 2012*). As a consequence, many modeling studies focussed on the extracellular fields induced by postsynaptic currents on the dendrites and the soma of a neuron (*Holt and Koch, 1999*; *Einevoll et al., 2013*; *Lindén et al., 2011*; *Lindén et al., 2010*; *Fernández-Ruiz et al., 2013*). However, a number of recent data analyses and modeling efforts have revealed that active, non-synaptic membrane currents can play an important role in generating population-level EFPs (*Reimann et al., 2013*; *Anastassiou et al., 2015*; *Schomburg et al., 2012*; *Ray and Maunsell, 2011*; *Belluscio et al., 2012*; *Waldert et al., 2013*; *Ness et al., 2016*; *Scheffer-Teixeira et al., 2013*; *Reichinnek et al., 2010*; *Sinha and Narayanan, 2015*; *Taxidis et al., 2015*), including far reaching potentials detectable at the scalp (*Baker et al., 2011*; *Teleńczuk et al., 2015*). Currents from the axon are still thought to be so small as to be of minor importance for the EFP.

One of the reasons for the assumption that axonal currents contribute little to the EFP is that the far field of an action potential traveling along an idealized straight and long axon is quadrupolar, meaning that it decays faster with distance than synaptic sources, which are typically dipolar (*Nunez and Srinivasan, 2006*). Surprisingly, theoretical (*Tenke et al., 1993*) and experimental studies indicated that the EFP of axonal responses may also have a dipolar structure. For example, *Blot and Barbour (2014)* reported an EFP with a characteristic dipolar structure in the vicinity of cerebellar Purkinje cell axons; other studies (*Swadlow et al., 2002*; *Swadlow and Gusev, 2000*) showed that the axonal part of the EFP of thalamocortical afferents showed a polarity reversal associated with a dipolar source, and classical current source density studies found dipolar current distributions in axonal terminal zones in the cortex and the lateral geniculate nucleus, and attributed these to axons because of conduction velocities (*Mitzdorf and Singer, 1978*; *Mitzdorf, 1985*; *Mitzdorf and Singer, 1977*; see also *Tenke et al., 1993*). Here we introduce another experimental system and show a strong dipolar, axonal field potential in the auditory brainstem of the barn owl.

The discrepancy between the quadrupolar structure of EFPs generated by idealized axons, and the experimentally observed dipolar structure raises the question of how axons are able to generate dipolar field potentials. In this manuscript we show how dipolar far fields in the EFP of axons can be explained by the axons' anatomical structure. In particular, the branchings and terminations of axons in their terminal zone area deform the extracellular waveform (*Gydikov et al., 1986*; *Gydikov and Trayanova, 1986*; *Plonsey, 1977*) and can lead to a dipolar EFP structure. Axon bundles, sometimes called fascicles, exist throughout the peripheral and central nervous system and have such terminal zones (*Kandel et al., 2000*; *Hentschel and van Ooyen, 1999*; *Nornes and Das, 1972*; *Goodman et al., 1984*). The white matter of the mammalian brain can be viewed as an agglomeration of such fascicles (*Schüz and Braitenberg, 2002*). We therefore predict pronounced contributions of axon bundles to EFPs, which are neglected in current models.

In what follows, axonal contributions to the EFP are first investigated by a numerical model based on forward simulation (*Holt and Koch, 1999*; *Gold et al., 2006*). This first model includes a large-scale multi-compartment simulation (*Jack et al., 1975*; *Rall, 1959*; *Abbott, 1992*; *Hines and Carnevale, 1997*; *Hines et al., 2009*) of an axon population. We then outline the basic mechanisms by means of a second, analytically tractable, model of a generic axon bundle. Finally, we validate model predictions with data from multi-site in-vivo electrophysiological recordings from the barn owl auditory brainstem.

## Results

### Effects of axonal bifurcations and terminations on extracellular action potentials

To understand how the geometry of an axon affects the extracellular waveform associated with action potentials, we first numerically simulated single action potentials propagating along generic axons and calculated their contribution to the EFP (for details, see Materials and methods). This was done for five scenarios: quasi-infinite axons, terminating axons, bifurcating axons, axons that bifurcate as well as terminate, and axon bundles (*Figure 1*). We began by simulating a long axon, approximating an infinitely long axon following a straight line path, neither bifurcating nor terminating (*Figure 1A*). The extracellular action potential has the characteristic triphasic shape. As the action potential travels along the axon, the waveform is translated in time with the conduction velocity, but is otherwise unchanged. The triphasic shape is also present in the spatial arrangement of transmembrane currents at any given time, which is the reason for the quadrupolar EFP response traditionally assumed for axons.

There are two ways of understanding the triphasic shape of the extracellular waveform. One way is by attribution of the peaks of the response to individual current types. The first, small positive peak corresponds to the capacitive current, the large and negative second peak to the sodium current, and the final positive peak to the potassium current (*Gold et al., 2006*). Another, more mathematical way of understanding the triphasic shape is specific to the nature of the axon. Due to Kirchhoff's law and cable theory (see Materials and methods), the local transmembrane current in a homogeneous axon is proportional to the second spatial derivative of the membrane potential along the direction of the axon. Because the action potential is roughly a traveling wave, the currents are

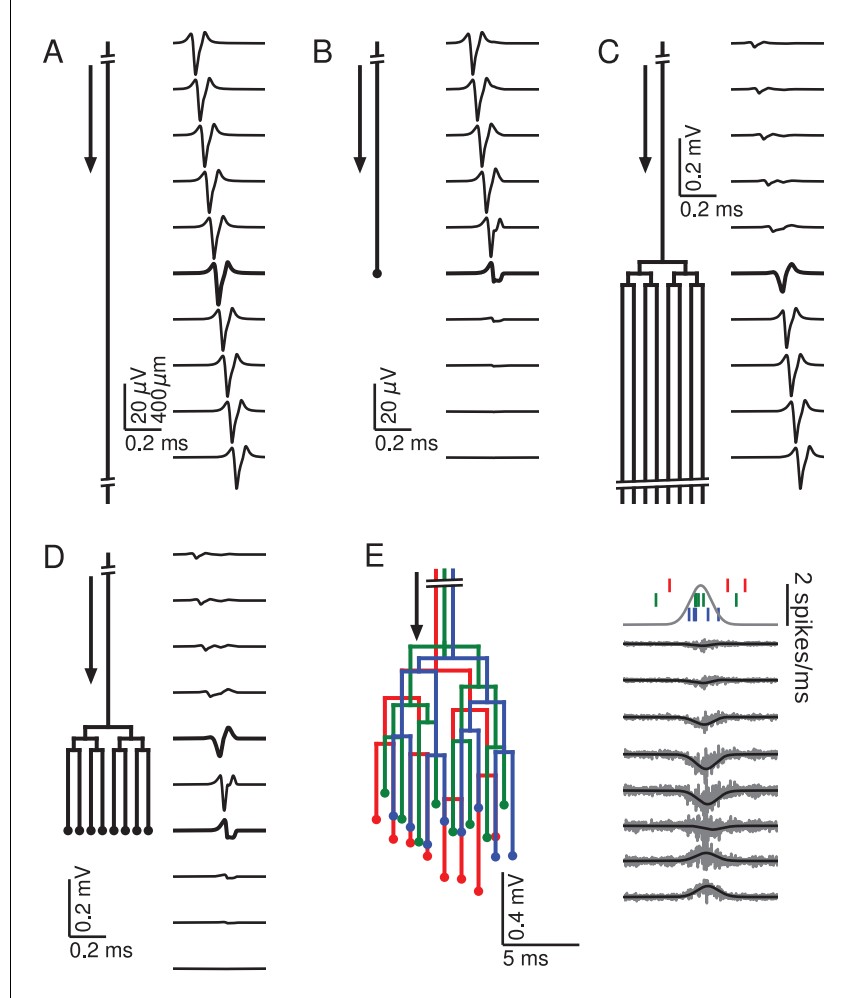

**Figure 1.** Relationship between axon morphology and extracellular potential. Multi-compartment simulations of action potentials traveling along axons with varying morphologies, as indicated by the diagram on the left-hand side of each subfigure. Action potential propagation direction indicated by arrow. Waveforms, shown on the right-hand side of each subfigure, were recorded at a horizontal distance of 150 µm from the axons. The vertical depth is indicated by the plot position, spaced by 400 µm. Horizontal plot location and distances between axons are for illustration only, all axons were simulated to lie on a straight line. (A) Action potential in a quasi-infinitely long, straight axon. (B) Terminating axon. Action potential waveform closest to the termination thickened for emphasis. (C) Branching axon. The axon branches multiple times within of 200 µm. Thicker waveform at the center of the bifurcation zone. (D) Combined bifurcations and terminations. Note the larger voltage scales in C and D, which correspond to the different number of fibers. (E) Response in a population of 100 randomized morphologies, three of which are shown schematically (colored). Activity consists of spontaneous background activity (100 spikes/s) superimposed with a brief Gaussian pulse of heightened spike rate (2000 spikes/s). Spike rate and example spike times for the three morphologies are shown at the top. Right: gray lines show activity of full population averaged over 40 trials, while the black lines show the low-pass (<1 kHz) component. Note that the time and voltage scales are different from A-D. In all graphs, spatial scales are the same, as indicated by the scale bar in A.
DOI: https://doi.org/10.7554/eLife.26106.002

also proportional to the second *temporal* derivative of the membrane potential. The three extrema of the EFP are thus related to the points of maximum curvature in the action potential waveform, namely the onset, the maximum, and the end of the spike.

Next we simulated the response of an axon that terminates (*Figure 1B*). Here the action potential approaching the recording location (top traces) has the same, triphasic EFP response as in the non-branching case. When the action potential reaches the termination point, its EFP gradually deforms

into a biphasic response, with a positive peak preceding a negative peak. The mechanism for this deformation can be understood as follows: As the action potential approaches the recording location next to the termination, the majority of the transmembrane currents flow at points located before the termination, and they are almost identical to those in the non-terminating case; the first, capacitive peak is not affected. As already mentioned, the second and third peaks of the extracellular action potential in the non-terminating case are generated by currents close to or after the electrode location. In the terminating axon, there are no currents at points after the termination, leading to a partial suppression of the second peak and a complete suppression of the third peak.

Another generic structure found in axons is a bifurcation. To emphasize the impact of bifurcations, we simulated a single axon that bifurcates three times on each branch within a distance of 200 µm (100 µm between branchings), leading to a total number of 8 collaterals leaving the bifurcation zone (*Figure 1C*). (Note that in order to avoid confounding effects, the horizontal distances between axons in *Figure 1C–E* are for illustration only; all collaterals were simulated to lie on a straight line.) The EFP far away from the bifurcation zone has a triphasic shape and resembles the one observed in *Figure 1A*, and the amplitude is proportional to the number of axon fibers. The EFP near the bifurcation zone has a biphasic shape. Although there is an initial tiny positive peak, the response is dominated by the second, negative and the third, positive peak. This waveform can again be understood by comparison to the first example (*Figure 1A*) containing the infinitely long axon: The tiny positive initial peak resembles the infinite case, because it is constituted by the action potential-related currents flowing within the part of the axon before the bifurcation. As the action potential passes the bifurcation zone, there are now several action potentials (one in each fiber). Because of the active nature of the action potential, the active currents are the same in each outgoing fiber as in the incoming fiber. This leads the second and third peak to be multiplied in size, yielding a quasi-biphasic response. We chose to simulate several bifurcations because this leads to a clearer effect in the EFP. In the case of a single bifurcation, this effect is also present, but the amplification of the second and third peak relative to the first peak is not as notable as in this example.

To understand how bifurcations and terminations interact when they are present in the same axon, we simulated an axon with an identical number of bifurcations as in the previous case, but then added terminations to all the fibers 700 µm after the bifurcation zone (*Figure 1D*). We found that this configuration leads to the same biphasic responses as observed in the cases of the isolated anatomical features. A triphasic response occurred in-between the bifurcation and termination zones. A notable point here is that the potential at the bifurcation and termination are both biphasic and on the same timescale, but opposite in polarity.

After having studied the EFPs of single axons, we next started to simulate axon bundles, because axons often run in parallel bundles in the brain. Moving towards more biologically plausible axon geometries, we considered bundles consisting of axons with slightly altered spatial arrangement: we randomly perturbed the precise locations of bifurcations and terminations in the axon tree (*Figure 1E*, left; for details, see Materials and methods). We simulated 100 axons and stimulated each axon with an inhomogeneous Poisson spike train (*Kuokkanen et al., 2010*; *Softky and Koch, 1993*). The driving rate of the inhomogeneous Poisson process was the same for all axons and consisted of a constant background rate (100 spikes/s) and a Gaussian pulse of heightened activity (2000 spikes/s). The standard deviation of the pulse was 1 ms, resulting in an additional 3.5 spikes per axon being fired on average over the entire duration of the pulse. The resulting extracellular population-level waveforms (*Figure 1E*, right) showed a polarity reversal reminiscent of *Figure 1D*. However, in the bifurcation zone, the summed contribution from many fibers and action potentials lead to a monophasic negative peak, and in the termination zone there was a monophasic positive peak. Interestingly, the summed potential at the center of the terminal zone largely cancelled out.

The fact that the responses in *Figure 1E* were mostly monophasic can be explained by the presence of a non-zero bias in the biphasic responses observed for the single spike responses in *Figure 1D*: close to a bifurcation, the area under the negative part of the curve slightly exceeded that of the positive part, and vice versa close to a termination. When summed up over many spikes with different timings, this difference in areas induced a positive or negative polarity of the population response in *Figure 1E*.

The reversal behaviour shown in *Figure 1E* is similar to the polarity reversal associated with a dipole observed in experimental studies (*Swadlow and Gusev, 2000*; *Swadlow et al., 2002*; *Blot and Barbour, 2014*). To summarize, simple one-dimensional model axon structures can

produce complex and diverse spatiotemporal EFP responses, including monophasic, biphasic and triphasic waveforms, comparable to experimentally recorded responses.

## Axonal projections generate a dipole-like field potential

Dipole-like EFPs have a much larger spatial reach than quadrupolar-like EFPs, which are typically associated with axons (*Nunez and Srinivasan, 2006*). To further understand whether and how axons can generate a dipolar EFP, in *Figure 2* we turned to three-dimensional axon morphologies, in contrast to the one-dimensional case studied in *Figure 1* (for details, see Materials and methods). We thus simulated a parallel fiber bundle of 5000 axons that at first runs at a constant number of fibers without bifurcations and then reaches a terminal zone. Within this terminal zone, the fibers first bifurcate, which increases the number of fibers. Finally, as the axons reach the end of the terminal zone, they terminate and the number of fibers decreases to zero (example axon shown in *Figure 2A*). To model more closely the actual axonal structures found in nature, we included a radial fanning out of the branches as well as a more diverse set of morphologies with a variable number of bifurcations and terminations (for details, see Materials and methods).

The spiking activity of a generic axon bundle was simulated by a background spontaneous firing rate of 100 spikes/s and a short pulse of increased activity. We chose a Gaussian pulse with an maximum instantaneous rate of 2900 spikes/s and a standard deviation of 2.8 ms. Note that this high driving rate is only the instantaneous maximum, and the actual firing rate is limited by the refractory period following a spike. These numbers are motivated by the early auditory system of barn owls (*Köppl, 1997a*; *Sullivan and Konishi, 1984*; *Konishi et al., 1985*), where instantaneous spike rates of 3000 spikes/s occur in response to click stimuli (*Carr et al., 2016*). However, our approach is not limited to the auditory system (which would also require the introduction of the synchronization of the spike times to the auditory stimulation frequency, called phase locking). Instead, this pulse of activity could relate to various kinds of evoked activity in the nervous system, such as sensory stimulation, motor activity or a spontaneous transient increase in population spiking activity.

To characterize the spatiotemporal dynamics of the evoked EFP, the time course of the potential was calculated for several locations along the axon trunk. The responses were averaged over 10 repetitions. We divided our analysis into two frequency bands by filtering the responses. The first frequency band was obtained by low-pass filtering with a cutoff frequency of 1 kHz (*Figure 2A–C*). The second frequency band was the multiunit activity (MUA) obtained by high-pass filtering with a cutoff frequency of 2.5 kHz (*Figure 2D–F*). To make the MUA easier to interpret in terms of overall activity reflected, it was half-wave rectified and low-pass filtered (<500 Hz, see Materials and methods). The two frequency bands showed a qualitatively different spatiotemporal response in the vicinity of the projection zone, as we will show in the following.

We first studied the effect of the Gaussian rate pulse on the low-pass filtered EFP (*Figure 2B*). The filtering removed most of the identifiable components of individual spikes, while a population-level signal remained, similar to an LFP signal. The distribution of the maximum amplitudes of these responses is shown by the colored contour lines in *Figure 2A* and the colored voltage traces in *Figure 2B*. Surrounding the terminal zone of the axon bundle in *Figure 2A*, low-pass filtered EFP amplitudes showed a double-lobed shape typical of a dipole.

In *Figure 2B*, the low-pass filtered EFP responses mostly showed monophasic deflections elicited by the population firing rate pulse, in a manner similar to that observed in *Figure 1E*. Such deflections were visible at all recording locations. In the radial direction away from the axon tree, i.e. in the horizontal direction in the figure, the low-pass filtered EFP amplitude decays. In the axial direction along the axon tree, i.e. in vertical direction in the figure, the voltage deflection reverses polarity in the middle of the terminal zone of the bundle (*Figure 2B*). The polarity reversal occurs by a decrease of the amplitude to zero and a subsequent reappearance with reversed polarity (as opposed to a polarity reversal through a gradual shift in phase). This behaviour is also typical for a dipolar field potential.

The point of the polarity reversal coincides with the middle of the terminal zone. Interestingly, this means that the absolute value of the response amplitude reaches a local minimum just at the axial location at which the number of axonal fibers reaches a maximum. To better understand how the anatomical features of the axon bundle and the EFP response amplitude are related, we compared its signed maximum value (meaning the *signed* value corresponding to the maximum *magnitude* of the amplitude of the EFP, black line in *Figure 2C*) with the change of the number of nodes

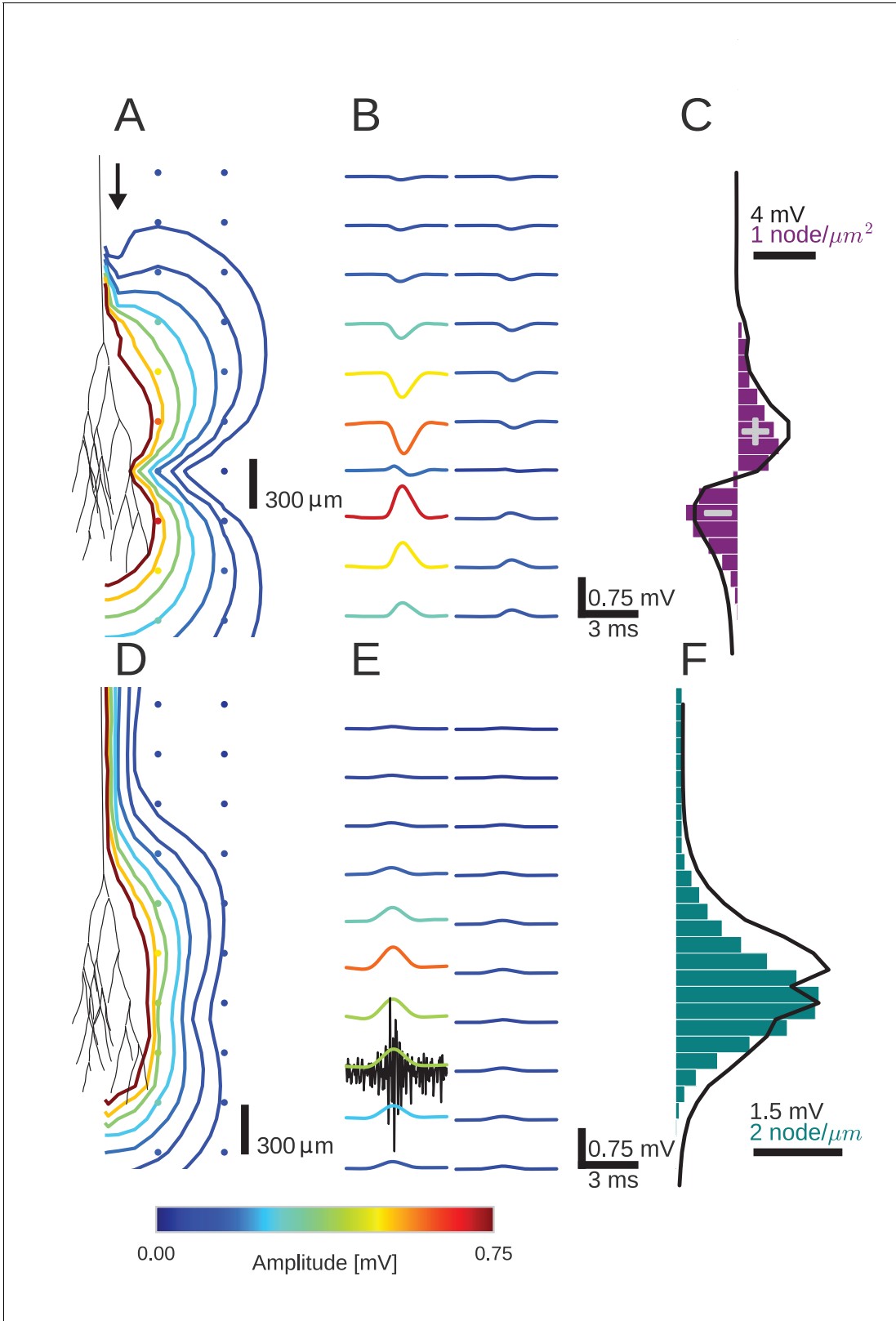

**Figure 2.** An activity pulse in an axonal projection generates a dipole-like extracellular field potential (EFP). (**A**) Modeled example axon from the simulated bundle in black, along with iso-potential lines for the low-pass filtered (<1 kHz) EFP signature of the activity pulse. The contours (amplitudes in mV as indicated by colorbar) show the typical double-lobe of a dipole. (**B**) The low-pass filtered EFP waveforms, recorded at the locations of the colored dots in *A*, show mostly unimodal peaks. The peak amplitude reverses polarity as a function of recording location in the vertical direction. The *Figure 2 continued on next page*

*Figure 2 continued*

reversal occurs by inverting the amplitude with approximately unchanged shape. (**C**) Progression of the maximum low-pass filtered EFP amplitude with depth (black line) at a distance of 100 μm from the trunk (indicated by arrow in A). The amplitude closely follows the local change (spatial derivative) in number of nodes per unit length (purple histogram), which is proportional to the difference in number between bifurcations and terminations. (**D**) Modeled axon from bundle as in A, and iso-potential contours for the multi-unit activity (MUA) component. (**E**) Response waveforms of the MUA component. High-pass filtered (>2.5 kHz) component (the first processing stage for calculation of MUA, see Materials and methods) in black. (**F**) Maximum amplitude of the MUA component (black line) follows the number of fibers (teal-colored histogram). Note the different units of the histograms in (**C**) and (**F**), due to the fact that (**C**) is the derivative in space of (**F**).

DOI: https://doi.org/10.7554/eLife.26106.003

per 200 μm bin (purple histogram in *Figure 2C*): Along the nerve trunk the number of fibers is constant. As the axon bundle reaches its terminal zone, the number of bifurcations increases (purple bars point to the right in *Figure 2C*). The increase of bifurcations is followed by an increase in terminations. In the middle of the terminal zone, the number of bifurcations and terminations are equal. At the same depth, the amplitude of the EFP component crosses zero. At the end of the terminal zone, the terminations outweigh the bifurcations (purple bars point leftwards in *Figure 2C*). As the axon bundle ends, there are no longer any bifurcations or terminations, and the number of fibers decays toward zero. Overall, the signed maximum amplitude EFP (black trace) follows the distribution of branchings and terminations (purple histogram). This progression of amplitudes in the low-frequency components seen in *Figure 2C* is also visible in *Figure 2B*, most clearly in the first column. The polarity reversal in the center of *Figure 2B* corresponds to the crossing of zero amplitude in *Figure 2C*.

To understand how the EFP contributions are related to individual spikes, we next turned our attention to the high-frequency MUA. The MUA is thought to reflect local spiking activity (*Stark and Abeles, 2007*). In *Figure 2D*, the iso-amplitude lines of the MUA appeared like an ellipsoid centered at the terminal zone (*Figure 2D*); they did not show the double-lobe shape observed for the low-pass filtered EFP in *Figure 2A*.

The shape of the MUA response was weakly dependent on the recording location. The main change across locations was in the scaling of the amplitude (*Figure 2E*). The amplitude decays with radial distance from the trunk. In the axial direction, the amplitude reaches its maximum in the middle of the fiber bundle. This dependence of the MUA amplitude on the axial location is further examined in *Figure 2F*. The amplitude of the MUA component (black trace) changes in accordance with the local number of nodes per unit length (teal-colored histogram), which is proportional to the number of fibers. The local number of fibers and the MUA amplitude are both constant along the nerve trunk. Both measures then increase in amplitude as the number of fibers is increased by bifurcations. As the fibers terminate and the number of fibers decreases, so does the amplitude of the MUA.

To conclude, we have shown a qualitatively different behaviour in the low- and high-frequency components of the EFP, i.e. for the low-pass filtered EFP and the MUA. The particular branching and terminating structure of the axon bundle may thus give rise to a dipolar low-pass filtered EFP.

## Effects of bifurcations and terminations on distance scaling of EFPs

To further demonstrate that bifurcations and terminations of axons give rise to a dipolar field, we investigated the effect of an axon terminal structure on the spatial reach of the EFP (*Figure 3*). Motivated by the fundamentally different spatial distributions of the low-pass filtered EFP and the high-frequency MUA in *Figure 2*, we again differentiated between these frequency bands and simulated an axon bundle containing a terminal zone with bifurcations and terminations. Moreover, as a control, we also simulated an axon bundle without bifurcations in which a fixed number of fibers simply terminates.

In order to separate the effects of any radial fanning out of the axon bundle from the effects of bifurcations and terminations, and to afford better analytic tractability, we transitioned back to a simpler one-dimensional model of the axon bundle (see Materials and methods). This model omitted the radial fanning out of the bundle in the terminal zone, as in *Figure 1*. Furthermore, we discarded the detailed conductance-based simulation of the membrane potential, and instead assumed a fixed membrane potential waveform traveling along the axon trunk with a constant propagation velocity. Using linear cable theory, it was then possible to calculate the membrane currents necessary for the

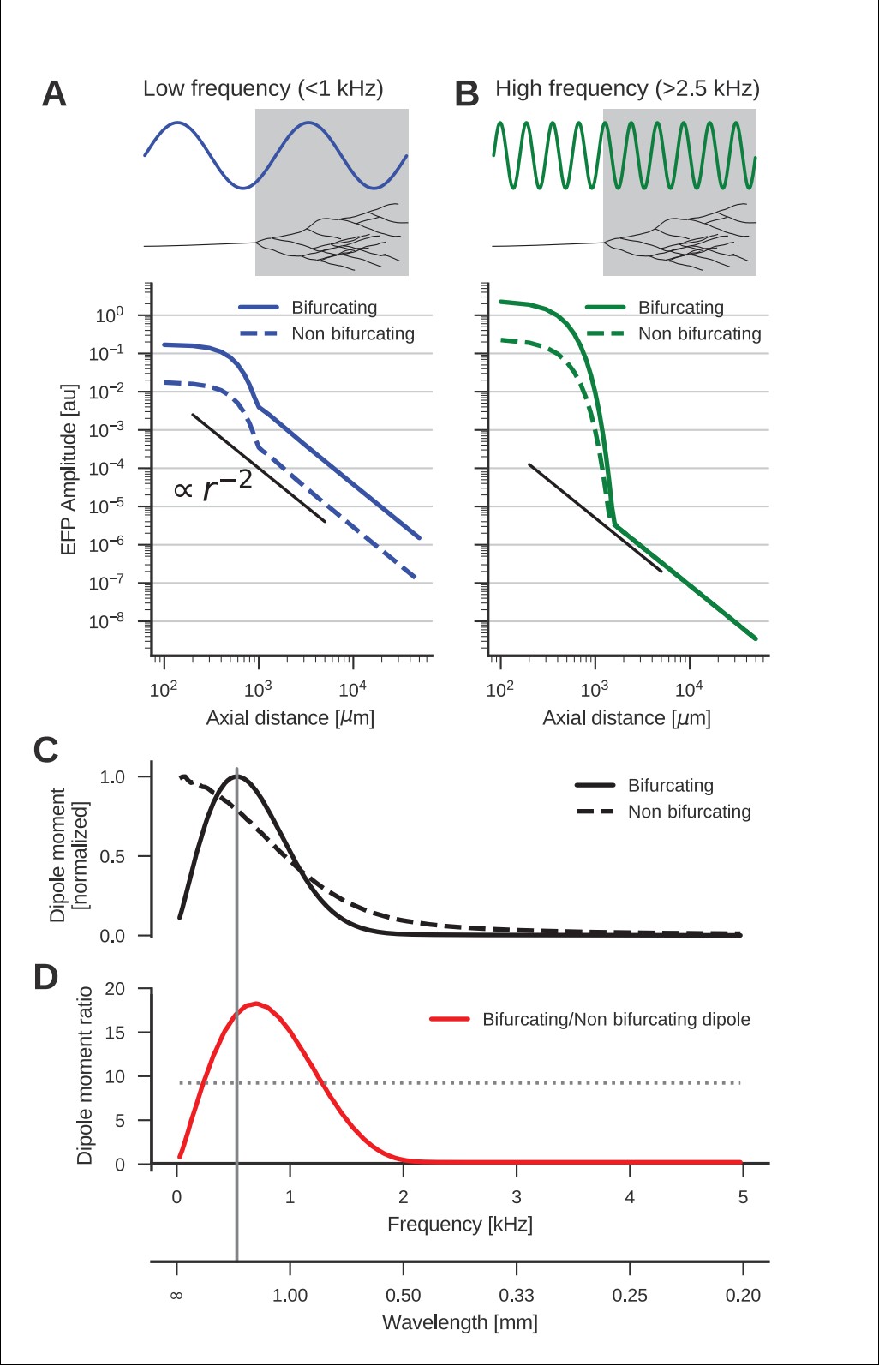

**Figure 3.** The low-frequency (<1 kHz) component of the axon bundle EFP is influenced supralinearly by a projection zone, while the high-frequency (>2.5 kHz) component is not. (**A**) Scaling of the low-frequency (<1 kHz) EFP component. (*Top*) The spatial wavelength of the membrane potential oscillation (blue) is larger than the width of the projection zone (gray). (*Bottom*) The amplitude of the low-frequency EFP component for the bifurcating

*Figure 3 continued on next page*

*Figure 3 continued*

case (solid line) decays with with axial distance from the axon bundle. It always exceeds the EFP amplitude of the non-bifurcating case (dashed line). Note the double-logarithmic scale. Axial distances $r$ are calculated from the center of the terminal zone. For comparison, scaling that follows $r^{-2}$ is indicated with the black line. (**B**) Same as A but for the high-frequency (>2.5 kHz) EFP component. (*Top*) The spatial wavelength of the membrane potential oscillation (green) is much smaller than the width of the projection zone (gray). (*Bottom*) The amplitude of the high-frequency EFP decays several orders of magnitude within the terminal zone, and the amplitude is larger in the bifurcating case (solid line) compared to the non-bifurcating case (dashed line). Far away from the terminal zone, i.e., for axial distances $r$>1 mm, they decay proportional to $r^{-2}$ but with similar amplitudes. (**C**) Normalized dipole moments of the bifurcating and non-bifurcating bundles as a function of frequency. (**D**) Ratio of the dipole moments between bifurcating and non-bifurcating cases (red line), compared to the maximum ratio 10 of the number of fibers (dotted line), to indicate supralinear (>10) and sublinear (<10) contributions. Vertical gray line in C and D indicates the width (~2 mm) of the projection zone.

DOI: https://doi.org/10.7554/eLife.26106.004

determination of the EFP. The analytic nature of the simplified model also allowed us to consider a continuous number of fibers instead of simulating discrete bifurcations and terminations. All following simulations are based on this simplified model.

To verify that this simpler one-dimensional model accurately captures the EFP response of an axon bundle, we applied parameters equivalent to those used in *Figure 2* and compared the resulting EFP to that obtained from the full biophysical model. We calculated the relative difference of the EFPs by taking the absolute value of their differences, and normalizing by the sum of their absolute values. Averaged over time, this relative difference at distances greater than 1 mm from the center of the projection zone was <0.05 in axial direction. In the radial direction, we found, as expected, larger relative discrepancies of <0.3 for radial distances > 1 mm. In what follows, we focus on the axial direction, which is the dipole axis.

Let us now specify how we simulated the two axon bundle morphologies. The control case was a non-bifurcating bundle, which had a constant number of 50 fibers up to the termination point, and then tapered out with a Gaussian profile that was centered at the termination point with a height of 50 fibers and width (standard deviation) of 300 μm. The second case was that of an axon bundle with a projection zone containing bifurcations. Here we added to the distribution of the number of axons used for the non-bifurcating control case a further Gaussian distribution to account for the projection zone. This additional Gaussian was also centered at the termination point, but had an amplitude of 450 fibers and a standard deviation of 500 μm, meaning that the overall width of the terminal zone was ≈2 mm. Unlike the tapering-out in the control condition, this component was added both before and after the termination point. It resulted in a maximal fiber number of 500 at the termination point, which is a factor 10 larger than in the control case. Both distributions constructed in this way were smooth, and they had smooth first derivatives in space. In both cases, the number of fibers decreased monotonically after the termination point. We considered a conduction velocity of 1 m/s in this example, though results are qualitatively the same for other values. In order to understand the frequency-specific effects of the projection zone, we calculated the responses to membrane potential components with temporal frequencies between 25 Hz and 5 kHz, with the same amplitude for each frequency component. For the conduction velocity 1 m/s, these temporal frequencies corresponded to spatial wavelengths from 10 mm to 0.2 mm, i.e. from much larger to much smaller than the width of the terminal zone. We then calculated for each frequency/wavelength the average amplitude of the resulting EFP response by taking its standard deviation. Due to the linear nature of our model, the frequency responses obtained in this manner are applicable to the Fourier components of any membrane potential time-course.

The dipole-like component observed in *Figure 2* for the low-frequency component had its dipole axis aligned with the axon trunk. We therefore considered the distance $r$ beyond the termination point in the direction extending the axon trunk, which we called the axial direction. Because we suspected a dipolar response, we expected the amplitude of the field potential to decay as $r^{-2}$. To test the scaling behaviour of this component, we first plotted the average amplitude of the low-frequency responses ($f$<1 kHz) in axial direction in *Figure 3A*. The plot is on a double logarithmic scale, meaning that the slope of the curve corresponds to the scaling exponent, and the vertical offset

corresponds to the amplitude of the size of the dipole moment, which is a measure for the strength of the dipolar EFP. We observed the expected $r^{-2}$ scaling for distances larger than the extent of the bifurcation zone ($\gtrsim 1$ mm).

Comparing the responses of the bifurcating axon bundle and the non-bifurcating control condition (full and dashed lines in *Figure 3A*), we saw that for short distances (<1 mm) the response of the bifurcating case was a factor 10 larger than the control. At these distances the response was due to the local fibers, of which there are 10 times more in the bifurcating case. At distances larger than 1 mm, we observed that the distance scaling was proportional to $r^{-2}$, meaning that there was a dipole moment in both conditions (a vanishing dipole moment would have implied a slope steeper than $r^{-2}$). Interestingly, for distances larger than 1 mm the response in the bifurcating case exceeded the control by a factor 20. We thus concluded that at low frequencies, the bifurcation zone contributes supralinearly to the dipole moment.

The reason for this supralinearity is that contributions from different parts of the axon bundle can interfere constructively or destructively. The maximum constructive interference occurs when the spatial width of the oscillation agrees with that of the projection zone (*Figure 3A*, top). Importantly, currents from fibers inside the projection zone on average interact destructively with those from outside the projection zone. The magnitude of this effect depends on the ratio of the number of fibers inside the projection zone compared to the number outside. The larger the ratio the smaller the impact of destructive interference. Thus, bifurcations suppress the destructive interference (*Figure 3A*, bottom, full line). On the other hand, for a ratio of one, i.e. the non-bifurcating case, destructive interference is strong, which diminishes the overall response amplitude (*Figure 3A*, bottom, dashed line).

Next, we examined the high-frequency (>2.5 kHz) component (*Figure 3B*). As in the low-frequency case, the response at distances < 1 mm was greater in the bifurcating case by a factor of 10. The asymptotic scaling was also $r^{-2}$ for axial distances >1 mm in both cases. However, unlike in the low-frequency case, the amplitudes were similar between bifurcating and non-bifurcating cases. Thus, the presence of a bifurcation zone did not contribute to the high-frequency dipole moments in the EFP. This feature is explained by the small spatial wavelength of the stimulus compared to the width of the bifurcation zone (*Figure 3B*, top)

## Frequency-dependence of dipolar axonal EFPs

We showed that the dipole moment depends on both the anatomy, that is, the presence of a projection zone, and the temporal frequency range (low vs. high frequencies) of the underlying activity. This relationship can be qualitatively understood by considering that in an axon bundle a voltage waveform propagates at some conduction velocity. The temporal frequency of this signal thus corresponds to a spatial frequency. If the spatial frequency of the membrane potential matches the width of the projection zone, the dipole moment can reach its maximum. In this case, at some point in time, positive membrane currents flow in one half of the projection zone and negative membrane currents flow in the other half (*Figure 3A*, top). For example, if the voltage waveform has a temporal frequency of 1 kHz and propagates at 1 m/s, the spatial wavelength is 1 mm. If the spatial width of the termination zone is about 1 mm, the dipole moment is maximal. In contrast, if the spatial wavelength is much smaller than the width of the projection zone, an alignment between projection zone and current flow is not possible, and the dipole is not amplified (*Figure 3B*, top) (for a detailed derivation see Materials and methods).

To quantitatively understand the frequency-specific contributions to the dipole moments, we examined the scaling behaviour of the EFP as a function of frequency. The amplitude of the dipole moment was determined by fitting a straight line with slope $-2$ to the double logarithmic scaling of the standard deviation of the EFP at a given frequency. The fit was performed for distances >1 mm. The extrapolation of this straight line to the axial distance 1 μm was then proportional to the dipole moment.

The normalized frequency-specific dipole moments are shown in *Figure 3C*. The dipole moment of the bifurcating case (solid line) has a maximum at around 500 Hz, as expected due to the agreement of the spatial wavelength (1 m/s / 500 Hz = 2 mm; gray vertical line in *Figure 3D and C*) and axial width (about 2 mm; gray box in *Figure 3A and B*) of the projection zone. The match is not exact because the shapes of sine wave and Gaussian are different. For lower and higher frequencies,

there is a mismatch in spatial wavelength and the width of the projection zone, meaning that the projection zone contributes less to the dipole moment, as also observed in *Figure 3B* for higher frequencies. In the non-bifurcating control case (dashed line) there is no projection zone, and the dipole moment decays monotonically with rising frequency because higher frequencies correspond to a smaller spatial separation of positive and negative currents, and thus to a smaller dipole moment.

In the bifurcating case, the maximum number of fibers was increased by a factor of 10. Accordingly, an increase in the dipole moment by a factor of 10 from the non-bifurcating to the bifurcating case could be explained by just linearly summing the dipole moments of individual fibers. An increase in the dipole moment by a factor greater than 10 would be supralinear. In *Figure 3D* we compared this relative impact of the terminal zone on dipole moments (red line) by plotting the dipole moment ratios across frequencies. The contribution of the terminal zone is greater than 10 (dotted line) for intermediate frequencies between about 200 and 1300 Hz, and smaller than a factor 10 outside this frequency range.

Together, these observations show us that the terminal zone makes a frequency specific contribution to the far-reaching dipole field potential of the axon bundle. This provides a deeper understanding of the findings of *Figure 2*: At low frequencies (<1 kHz), we observed a supralinear dipolar behaviour due to the specific morphology the bundle, with the projection zone forming the dipole axis. At higher frequencies, the bifurcation zone does not amplify the dipole moment, meaning that we could observe responses mainly locally.

## The barn owl neurophonic potential in nucleus laminaris as an example for a dipolar field in an axonal terminal zone

To test our prediction of dipolar extracellular field potential responses due to axon bundles, we recorded EFP responses from the barn owl auditory brainstem. The barn owl has a highly developed auditory system with a strong frequency-following response in the EFP (up to 9 kHz, [*Köppl, 1997b*]), called the neurophonic, which can be recorded in the nucleus laminaris (NL). In NL, the input from the two ears is first integrated to calculate the azimuthal location of a sound source, and this information is encoded in the EFP (*Carr and Konishi, 1990*). The EFP in this region is mainly due to the afferent activity, and the contribution of postsynaptic NL spikes is small (*Kuokkanen et al., 2010*; *Kuokkanen et al., 2013*). Furthermore, the anatomy of the afferent axons is well known and follows a stereotypical pattern (*Carr and Konishi, 1988*; *Carr and Konishi, 1990*): Two fiber bundles enter the nucleus, with fibers from the contralateral ear entering ventrally, and from the ipsilateral ear entering dorsally. The axon bundles reach the NL from their origin without bifurcating, then bifurcate multiple times at the border of the NL, and then terminate within NL. Axon bundles have a strong directional preference and run roughly in parallel. Most of the volume within NL consists of incoming axons. This well studied physiology and anatomy makes the system an ideal candidate to investigate the EFPs of axon bundles; see the Discussion for arguments why synaptic contributions to the EFP could also be neglected here.

To explore the spatiotemporal structure of the EFP in NL, we performed simultaneous multi-electrode recordings of the response in NL (*Figure 4A*) to contralateral monaural click stimuli. The click responses showed distinct low-frequency (*Figure 4B*) and high-frequency (*Figure 4C*) components, as previously reported (*Wagner et al., 2009*). The frequency of the high-frequency ringing corresponds to the recording location on the frequency map within NL, and the ringing reflects the frequency tuning and phase locking of the incoming axons. In addition, there is a low-frequency component in the response (*Figure 4B*). We filtered the data to separate these components, using the same cutoff frequencies as before for low-frequency (<1 kHz) and high-frequency (>2.5 kHz) EFP.

The same simplified model used in *Figure 3* was fit to the data (example shown in *Figure 4*) by performing a nonlinear least squares optimization. The model considered only the average membrane potential across the fibers, and we calculated the membrane currents based on the density of fibers instead of simulating individual fibers. The model also discarded the radial extent of the bundle, treating it as a line; see Materials and methods for more details on the model. Free parameters to be fit were (1) the number of fibers at the depth of each recording location, (2) the average spatial derivative of the membrane potential over time in the fibers at the location next to the topmost, dorsal electrode, (3) the axonal conduction velocity, and (4) the average distance between the axon bundle and electrode array.

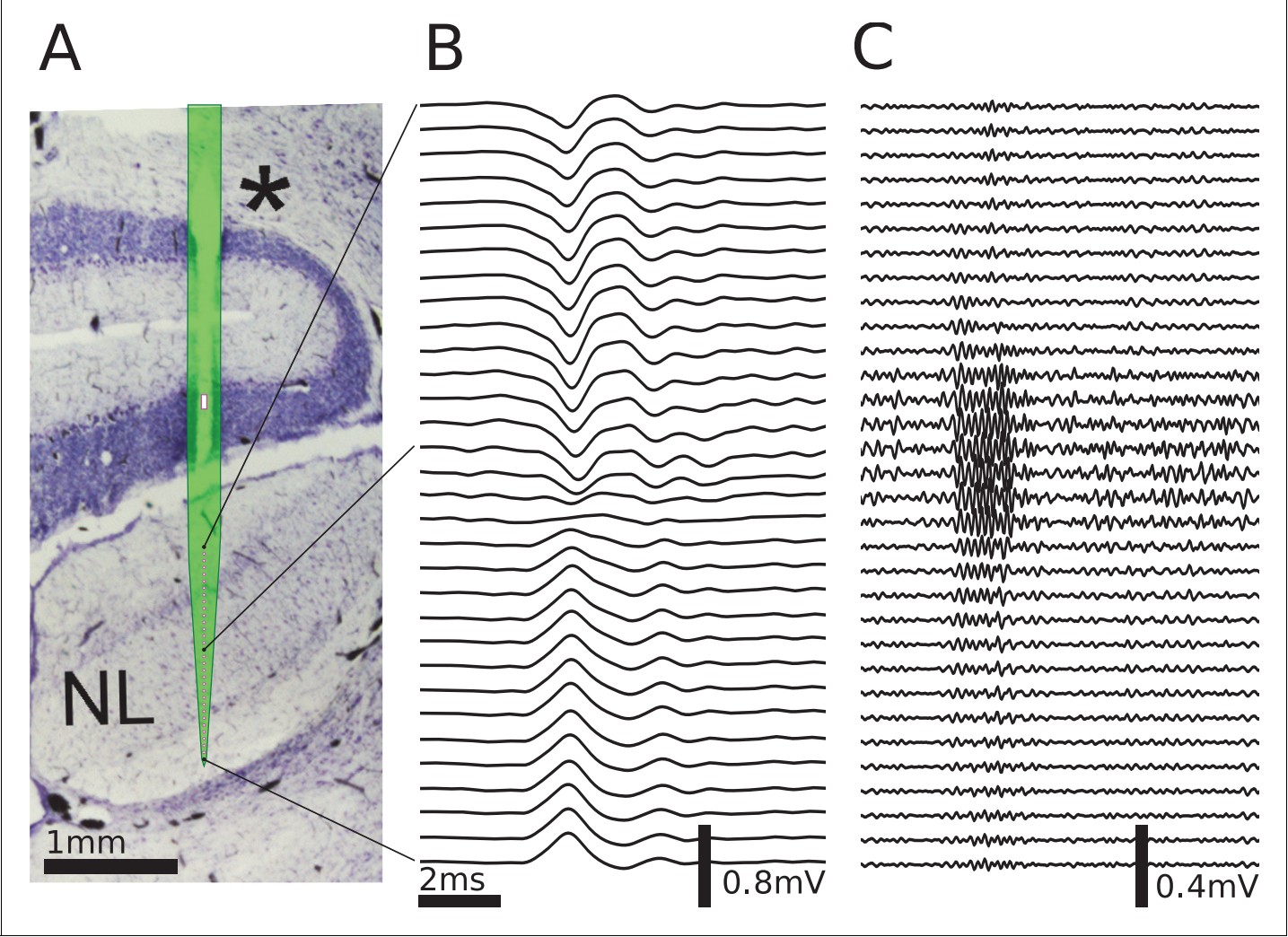

**Figure 4.** Multielectrode recordings in the barn owl show dipolar axonal EFPs. (**A**) Photomicrograph of a 40 µm thick transverse Nissl stained section through the dorsal brainstem, containing a superimposed, to scale, diagram of the multielectrode probe. The probe produced a small slit in a cerebellar folium overlying the IVth ventricle (*), and penetrated into the nucleus laminaris (NL). The recordings were made in NL, and electrodes extended to both sides of the nucleus. The outline of the probe is shown in light green, with the recording electrodes indicated by magenta dots, and the reference electrode as a magenta rectangle. The low-frequency (<1 kHz) component (**B**) and the high-frequency (>2.5 kHz) component (**C**) are ordered in the same way as the electrodes, with three examples connected to their recording sites by black lines. The time scales in B and C are identical (indicated by scale bar). Traces were averaged over 10 repetitions. Voltage scales are indicated by individual scale bars.

DOI: https://doi.org/10.7554/eLife.26106.005

The resulting EFP responses and the model fit are depicted in *Figure 5*. *Figure 5A* shows the inferred average over trials of the deviation of the membrane potential from the resting potential in response to the stimulus, at a location in the axon next to the first electrode (penetration depth 1550 µm), obtained from the fit. The inferred voltage is composed of high- and low-frequency components similar to those observed in the EFP. The inferred number of fibers as a function of dorso-ventral depth is shown in *Figure 5B*. The number (scaled by an arbitrary factor) has its maximum at the center of the electrode array, and decays steadily to both sides. This profile of the number of fibers is consistent with the known anatomy of axons in NL (*Carr and Konishi, 1990*; *Kuokkanen et al., 2010*).

The low-frequency (<1 kHz, *Figure 5C*) responses reveal the typical polarity reversal that we predicted for an axonal terminal zone (*Figures 1* and *2*). The dorsoventral depth in *Figure 5C and D* is

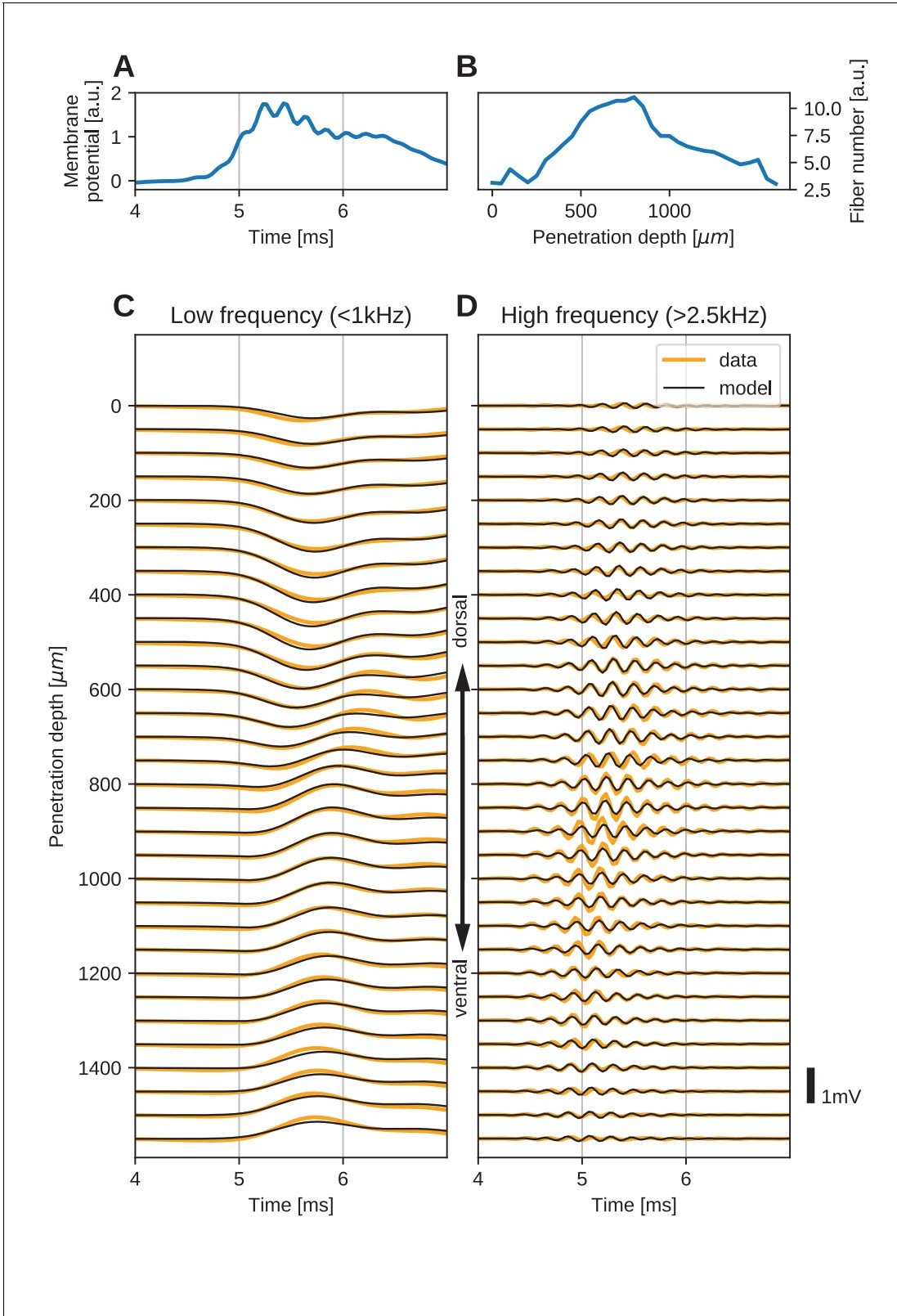

**Figure 5.** The spatial structure of EFPs recorded from the nucleus laminaris of the barn owl can be explained by a model of axonal field potentials (for details, see Materials and methods). (**A**) Membrane voltage, averaged across fibers, in the model when fit to the data. (**B**) Fitted number of fibers in the model as a function of penetration depth. (**C**) Low-frequency (<1 kHz) components of the EFP in response to a click stimulus at time 0 ms, at different

*Figure 5 continued on next page*

*Figure 5 continued*
recording depths. The depth is measured in the direction from dorsal to ventral. Recorded responses (orange) are shown along with model fits (black).
(**D**) High-frequency (>2.5 kHz) responses in recordings (orange) and model (black). Recorded traces were averaged over 10 repetitions.
DOI: https://doi.org/10.7554/eLife.26106.006

on the vertical axis, which corresponds to the horizontal axis in *Figure 5B*. The orange lines indicate the actual responses in the data.

The low-frequency responses at the dorsal and ventral edges in *Figure 5C* show the same shape, but with opposite polarity, as expected for a dipolar field. Note that for a pure dipole field, the amplitude of the central responses have zero amplitude. In the data shown here, central responses show a diminished maximum amplitude, which we interpret as the contribution of higher-order (mostly quadrupole) components. The model is able to capture the behaviour of this quadrupolar component as well, with a slight underestimation of the amplitude of the peak at ventral locations. The model even captures a small oscillation in the data with period of $\approx 1$ ms in the center of the recording. Here, too, the small deviations are likely due to slightly inhomogeneous conduction velocities or non-axonal sources.

In addition to the dipolar behavior of the low-frequency response, we also examined the high-frequency (>2.5 kHz) response, shown in *Figure 5D*. The responses have a Gabor-like shape, as expected (*Wagner et al., 2009*), with maximum amplitude in the center of the recording array, at around 850 µm penetration depth. The axonal conduction velocity was calculated to be 4.0 m/s, and the distance from the bundle was 162 µm. A previously published estimate of the axonal conduction velocity in this nucleus (*McColgan et al., 2014*) gave a confidence bound of 0.4–6 m/s. Toward the edges (<100 µm and >1400 µm), the amplitude decays. In the central region (400–1200 µm recording depth), a systematic shift in delay can be observed, while the response appears stationary in the more dorsal and ventral electrodes. The delay increases from ventral to dorsal, which is consistent with the anatomy for contralateral stimulation.

All these aspects of the data are qualitatively reproduced by the model (*Figure 5C and D*, black traces). The main deviation between model and data lies in a diminished amplitude of the high-frequency oscillation modelled at the most central electrode sites (*Figure 5D*). Because the phase shift in the central region is mainly determined by the conduction velocity, this mismatch might be due to a variable conduction velocity in the nucleus, and the constant velocity in the model. *McColgan et al. (2014)* showed that different conduction velocities exist in the core and periphery of the nucleus, as predicted by *Carr and Konishi (1990)* from variable internode distances. A diminished amplitude in the fit could reflect an inability of the model to exactly match the phase progression. Another possible explanation is that the additional amplitude could be due to non-axonal sources such as synaptic currents or postsynaptic spikes, which do not follow the assumptions underlying our model; see the Discussion for arguments why we expect such contributions to the EFP to be small.

When comparing the inferred membrane potential response (*Figure 5A*) to the measured EFP response (*Figure 5C and D*), the most salient difference is the dissimilar sizes of the frequency components. In the EFP, the low-frequency component has a comparable amplitude to the high-frequency component, but in the membrane potential the low-frequency component is much larger. This is because the EFP is related to membrane currents, which are proportional to the first and second derivatives of the membrane potential, and taking the derivative is equivalent to applying a high-pass filter.

We performed the fitting procedure (example in *Figure 5*) for 26 recordings from 3 different owls, with monaural stimulation from both ears (implying the activation of distinct axonal populations). The average correlation coefficient for all recordings was $R^2 = 0.56 \pm 0.15$. The correlation coefficient for the example shown in *Figure 5* was 0.62.

## Dipole moments of idealized axon bundles

We have shown theoretically and experimentally for specific examples of axonal projection zones and inputs how dipolar EFPs emerge. We now generalize this approach to predict the resulting dipolar EFP for arbitrary axon and stimulus configurations. Based on our cable-theory model, we

analytically derived the maximal dipole moment $p_{\max}$ for a large range of scenarios. From a given dipole moment the maximum far field potential at distance $r$ can be calculated as $\phi_{\max} = \frac{p_{\max}}{4\pi\sigma_e r^2}$ where $\sigma_e$ is the extracellular conductivity.

To simplify the analytical derivation as much as possible, we assumed a Gaussian waveform for the membrane potential of a single spike, with an amplitude $\bar{V}_{\text{spike}}$ and a width $\sigma_{\text{spike}}$. The resting membrane potential was irrelevant because only the first and second derivatives of the membrane potential contribute. The axon bundle population consisted of fibers with radius $a$, axial resistance $r_L$, and conduction velocity $v$. The population was assumed to be driven with a Gaussian firing-rate pulse with maximum firing rate $\bar{\lambda}_{\text{pulse}}$ and width $\sigma_{\text{pulse}}$. The distribution of the number of fibers at a given depth location was also described with a Gaussian, with width $\sigma_n$ and maximum number $\bar{n}$. This is an adequate approximation if the spikes in the incoming fibers contribute little to the dipole moment before reaching the projection zone. In this scenario, we calculated the peak dipole moment of the bundle (see Materials and methods for details) to be

$$p_{\max} = \frac{2\pi^2 a^2 \bar{n} \bar{\lambda}_{\text{pulse}} \bar{V}_{\text{spike}}}{\sqrt{e} r_L} \cdot \frac{v \sigma_n \sigma_{\text{pulse}} \sigma_{\text{spike}}}{\left(\sigma_n^2 + v^2 \left(\sigma_{\text{pulse}}^2 + \sigma_{\text{spike}}^2\right)\right)} \quad . \tag{1}$$

*Equation 1* tells us that the dipole moment is proportional to $a^2$, $\bar{n}$, $\bar{\lambda}_{\text{pulse}}$, $\bar{V}_{\text{spike}}$, and $1/r_L$. The dependence on $v$ and the widths is more complicated; the response is maximal with respect to the three (spatial) widths $\sigma_n$, $v\sigma_{\text{pulse}}$ and $v\sigma_{\text{spike}}$ when they satisfy the condition $w_1^2 = w_2^2 + w_3^2$ where $w_1$ is the largest of the three terms, while $w_2$ and $w_3$ are the other two terms, regardless order. The dipole moment is thus maximal when the widths of the spike, the pulse, and the terminal zone agree. In particular, if $\sigma_n$ (the width of the terminal zone) is the widest, then the dipole moment is maximal if $\sigma_n$ is equal to the spatial width of the overall activity in the axons, which is $v\sqrt{\sigma_{\text{spike}}^2 + \sigma_{\text{pulse}}^2}$. The widths add in this way because the overall activity is the convolution of two Gaussians.

Using this formula, it is then possible to calculate the expected contributions to the EFP for different scenarios. To test the approximation in the case of the barn owl, we chose the following values: axon radius $a$ = 1 µm, conduction velocity $v$ = 4 $\frac{\text{m}}{\text{s}}$ as inferred in the previous section, axial resistivity $r_L$ = 1 $\Omega$m, and extracellular conductivity $\sigma_e$ = 0.33 $\frac{\text{S}}{\text{m}}$ as used in studies of the cortex (*Gold et al., 2006*; *Holt and Koch, 1999*), anatomical and physiological parameters $\sigma_n$ = 500 µm, $\bar{n}$ = 80000, $\bar{V}_{\text{spike}}$ = 70 mV from (*Carr and Konishi, 1990*), and activation patterns for click stimulation from (*Köppl, 1997a*; *Carr et al., 2016*): $\bar{\lambda}_{\text{pulse}}$ = 1000 spikes/s, $\sigma_{\text{spike}}$ = 250 µs, $\sigma_{\text{pulse}}$ = 0.5 ms. This leads to a value for the dipole moment of $p_{\max} \approx 2.5\,\mu\text{A}\cdot\text{mm}$. At a distance of 750 µm, roughly the furthest distance recorded with the multielectrode array in *Figure 4* and *Figure 5*, this dipole moment corresponded to a field potential of 1.1 mV, consistent with the order of magnitude of the responses in our experiments (*Figures 4* and *5*).

Dipole sources are also to be expected to make up the majority of the electrical signals recorded at the scalp (*Nunez and Srinivasan, 2006*). One such signal is the auditory brainstem response (ABR), which is recorded at the scalp in response to auditory stimulation with clicks or chirps (*Riedel and Kollmeier, 2002*). An amplitude of about 10 µV of the ABR in the barn owl has recently been reported by *Palanca-Castan et al. (2016)*. We calculated the contribution expected from an axon bundle with the same characteristics as described before at 2 cm from NL, aiming to estimate the contribution to the ABR. Multiplying by a factor of 2 to account for the fact that there is an NL in each hemisphere, the predicted contribution was 3.1 µV, which is of the same order of magnitude as the value reported in the experiments.

To estimate the low-frequency dipole moment of NL from our multielectrode recordings, it is sufficient to use CSD analysis in one dimension, i.e. $\frac{\partial^2}{\partial z^2}\phi(z) = \frac{1}{\sigma_e}i(z)$ and a simple sinusoidal approximation of the voltage within NL: $\phi(z) = \phi_0 \sin(2\pi z/L)$ for $-L/2 < z < L/2$ and $\phi(z) = 0$ otherwise, where $\phi_0 \approx 0.5$ mV is the amplitude, $L \approx 2$ mm is the spatial wavelength, and $z$ is the depth in NL with $z = 0$ being in the center. To convert the current density $i$ into a current, we approximate the NL volume that contributes to the dipole as $V_{\text{NL}} \approx 6$ mm$^3$ (*Kuokkanen et al., 2010*). We assume that the current is homogeneously distributed in the directions perpendicular to $z$. Using the definition of a dipole, $p_{max} := \int \mathrm{d}V\, i(z)z$, we can integrate over the dimensions perpendicular to $z$ and obtain

$p_{max} = \frac{V_{\text{NL}}}{L} \int_{-L/2}^{L/2} \mathrm{d}z\, i(z)z$. Substituting $i(z)$ and solving the integral, we find the maximum dipole moment to be $p_{max} = 2\pi V_{\text{NL}}\sigma_e\phi_0/L \approx 3\ \mu\text{A} \cdot \text{mm}$, which is consistent with our previous estimates.

As a second example, we considered thalamocortical projections, for which (*Swadlow and Gusev, 2000*) reported amplitudes of extracellular spike-related potentials, called axon terminal potentials, at various locations; for example, at 400 µm from the center of the dipole, they reported an amplitude of the response of $\approx$ 1 µV. Individual thalamocortical axons are thin and have large and highly branched projection zones (*Feldmeyer, 2012*), so we estimated $\sigma_n$ = 250 µm, $\bar{n}$ = 30, and $a$ = 1 µm. We assumed a jitter $\sigma_{\text{pulse}}$ = 125 µs in the arrival time instead of a true activity pulse, and we normalized the pulse to have area 1 because we were considering a spike triggered average. The conduction velocity has been reported as $v$ = 8.5 m/s (*Simons et al., 2007*). Leaving all other values as in the previous approximation, we arrived at a dipole moment of $p_{\text{max}} \approx 1.5\ \mu\text{A} \cdot \mu\text{m}$, yielding an extracellular spike amplitude of $\approx$ 2.3 µV at the distance of 400 µm, which is of the same order of magnitude as the value ($\approx$ 1 µV) reported by *Swadlow and Gusev (2000)*.

In cases in which the jitter $\sigma_{\text{pulse}}$ is longer, the dipole moment is lower. For example, for pulses evoked by a visual stimulus, the pulse durations can exceed 10 ms (*Mitzdorf, 1985*; *Schroeder et al., 1991*; *Schroeder et al., 1998*; *Self et al., 2013*). Using the same parameters as for the thalamocortical projection employed before, but increasing the number of fibers by a factor of 100, increasing the width of the pulse to 10 ms, and increasing the maximal firing rate to 10 Hz, we found that the value of the dipole moment was $p_{\text{max}} \approx 0.018\ \mu\text{A} \cdot \mu\text{m}$, which is two orders of magnitude smaller than in the case of the brief pulse discussed above. However, when we further reduced the conduction velocity to 0.4 m/s, the same 10 ms pulse produced a dipole moment of $0.39\ \mu\text{A} \cdot \mu\text{m}$. Such low conduction velocities can, for example, be found in cortico-cortical projections (*Swadlow, 1989*).

To summarize, *Equation 1* quantitatively predicts the contribution of axonal projection zones to the far field EFP, and this prediction matched experimental values in several cases.

## Discussion

Numerical simulations, analytical calculations, and experimental data allow us to show how axonal fiber bundles may contribute to the EFP, and explain how the contributions are shaped by axonal morphology. There are three principal effects of axon bundle structure on the EFP. First, the low-frequency components of the EFP are governed by the densities of bifurcations and terminations and can have a dipolar structure (*Figure 1* and *Figure 2A–C*). Second, the high-frequency components are governed by the local number of fibers (*Figure 2D–F*). Third, membrane potentials that change on the same spatial scale as an axonal projection zone through which they propagate generate strong dipole moments in the EFP response (*Figure 3*). At the temporal frequencies that correspond to wavelengths of the size of the projection zone, this leads to dipolar EFP components that are not negligible and exceed the reach of the presumed quadrupolar nature of axonal EFPs.

### Relevance to the interpretation of electrophysiological recordings

Our findings relate to the interpretation of a wide range of electrophysiological data in general, and to the estimation of current sources in particular. When performing a typical current source density (CSD) analysis, the local number of fibers cannot be disentangled from membrane current density (*Nicholson, 1973*; *Potworowski et al., 2012*). In CSD analysis, the membrane current densities can vary independently with time and location. In contrast, in the case of axon fiber bundles, the situation is different: the number of fibers is variable in space, in particular in the terminal zone, but the current sources at different locations are highly correlated because they are caused by propagating action potentials. In the case presented here (*Figure 5*) where axonal action potentials dominate the EFP, it was possible to recover the (normalized) fiber densities and average membrane potentials from the recordings.

Beyond recovering actual fiber densities and membrane potentials, our approach enables the interpretation of CSD results in the presence of axon fiber bundles. For example, the sink and source distribution found in classical CSD analysis of axon bundles (*Mitzdorf and Singer, 1978*; *Mitzdorf, 1985*; *Mitzdorf and Singer, 1977*) shows a dipolar structure in terminal zones, but a conclusive explanation of their origin was not given. (*Tenke et al., 1993*) studied the dipole at an axonal

terminal zone in the macaque striate cortex for a fixed point in time, attributing the sinks to the depolarized axon endings, and the sources to the return currents distributed along the axons, while not taking account of additional currents flowing at bifurcations. Our modeling approach provides a novel way of interpreting these findings in terms of actively propagated action potentials in a fiber bundle.

As an example case for a fiber bundle, we recorded from the barn owl nucleus laminaris. *Figure 4* and *Figure 5* showed that the low- (<1 kHz) and high-frequency (>2.5 kHz) components exhibit qualitatively different behaviours as a function of recording location relative to the terminal zone. The low-frequency component is a largely stationary phenomenon, while the fine structure of the high-frequency component shifts gradually in space as a function of the axonal conduction velocity (*Figure 5*, see also (*Carr et al., 2015*)). Low-frequency components have a strong dipole moment, meaning that they contribute to the far-field EFP. Due to the difference in reach, the high-frequency component is most suitable for the study of local phenomena while the low-frequency component bears information about locations more distant from the recording site (*Figure 3*), consistent with findings on non-axonal EFPs (*Pettersen and Einevoll, 2008*; *Łęski et al., 2013*).

Note that the low-pass (<1 kHz) filtered EFP is calculated in a similar way to the LFP, with the exception that the cutoff frequencies used to separate the low- and high-pass filtered EFP are relatively high compared to those used in cortical or hippocampal studies to separate LFP and MUA. We applied these high cut-offs because our modeling and experiments were performed in the auditory brainstem of the barn owl, which operates on very short time scales and, consequently, higher frequencies. We expect other systems operating on slower time scales to have lower optimal cutoff frequencies separating the components. *Equation 1* indicates how the different spatial and temporal system properties relate to each other to generate a dipole moment.

Dipolar fields are essential for the generation of electrical field potentials at greater distances from the brain. The most prominent of these is the EEG, which is commonly attributed to the dipolar contributions of pyramidal cells (*Nunez and Srinivasan, 2006*). As originally suggested by *Tenke et al. (1993)*, we propose that axonal contributions might also be relevant in the analysis of these fields. This is particularly true for the auditory brainstem response (ABR), which is closely related to the EEG and involves brain structures that display high degrees of synchrony as well as axonal organization, and are thus ideal candidates for the generation of axonal field potentials visible at long ranges. This would in turn have implications for the interpretation of the ABR in clinical contexts.

The ABR amplitude of the barn owl has been reported to be on the order of 10 µV (*Palanca-Castan et al., 2016*) while we estimated a contribution of about 3 µV amplitude from the incoming axons in NL alone. This estimate of 30% axonal contribution to the ABR suggests that there may indeed be measurable components due to axons in the ABR, in particular, and the EEG, in general. However, this estimate is crude because it did not take into account the anatomy of the skull except for its size. Future studies based on a more detailed skull model and paired recordings of ABR and EFP should improve our understanding of axonal contributions to the ABR.

We have shown that the EFP in the barn owl NL is consistent with a model of axonal sources. We believe synaptic contributions to be small in this case for the following reasons: The somatic membrane potentials due to synaptic currents are much smaller than the impact of postsynaptic spikes (*Ashida et al., 2007*; *Funabiki et al., 2011*). Since postsynaptic spikes contribute little to the EFP (*Kuokkanen et al., 2010*), we suspect that the synaptic contributions to the EFP are also small. Furthermore, synaptic EFP contributions would require a spatial separation of currents, which is not possible to achieve in NL because of the symmetrical arrangement of synapses on the spherical NL cell bodies (*Carr and Konishi, 1990*), meaning that synaptic sources can also not explain a dipolar EFP, and are thus expected to contribute little to the EFP.

## Dipolar EFPs in other animals and brain regions

It is interesting to note that a similar dipole-like reversal of polarity as shown here for the barn owl NL has been reported in the chicken NL (*Schwarz, 1992*) as well as in the mammalian analog to NL, the medial superior olive (MSO) (*Mc Laughlin et al., 2010*). The phase reversal in this case was modeled based on the assumption that the postsynaptic NL and MSO dendrites with their bipolar morphology generate the dipolar EFP response (*Mc Laughlin et al., 2010*; *Goldwyn et al., 2014*). However, in the owl this dipolar morphology of neurons is largely absent (*Carr and Konishi, 1990*),

making dendritic sources unlikely. This differential morphology suggests that similar dipolar field potentials in owls and mammals emerge from different physiological substrates. Such a convergence might point towards an evolutionary pressure favoring a bipolar EFP structure in coincidence detection systems, and indeed, (*Goldwyn and Rinzel, 2016*) have proposed a model in which this extracellular potential enhances the function of coincidence detectors through a form of ephaptic interaction. Their approach centers on dendrites and is not directly transferable, but it seems possible that a similar mechanism might arise in the barn owl NL based on axonal EFPs.

The key assumption underlying our modeling of axonal geometries is that there exists a preferential direction of the axon arbor. In many structures this is the case, for example in the parts of the auditory brainstem we studied here. In other brain regions, this tendency is not as prominent, with a spectrum existing between completely directed and undirected growth. More undirected growth would lead to a more diffuse response in the EFP, and eventually to a cancellation of the dipolar field potentials. *Cuntz et al. (2010)* and *Budd et al. (2010)* studied the principles underlying the growth patterns of axons and found that the degree of direction in the growth of an axon depends on the balance struck between conduction delay and wiring cost. Optimizing for minimal conduction time leads to highly directed structures while optimizing wiring cost leads to more tortuous, undirected growth. This insight suggests that directed structures - and thus also strong, dipolar EFPs due to axons - may be more prevalent in systems which require high temporal precision in the information processing. This requirement for high temporal precision also aligns well with our model prediction: the dipole moment (*Equation 1*) is maximal when the spatial spread of the activation is equal to the size of the projection zone, favoring short activation times (<1 ms) for typical projection zone sizes (1 mm) and conduction velocities (1 m/s).

## Relationships to more detailed biophysical models

In the systems we were aiming to describe with our model, for example NL and thalamocortical projections, synaptic boutons are typically small, and we did not model them explicitly. In other systems such as the neuromuscular junction, the synaptic structure can be very large when compared to the axon bundle (*Harris and Ribchester, 1979*; *Katz, 1961*; *Katz and Miledi, 1965*). Such a large junction with an overall length of up to 1 mm was modeled by *Gydikov and Trayanova (1986)*. They found a significant effect of this structure on the EFP. The single flaring and tapering neuromuscular junction in their model had a comparable effect as the entire projection zone in our model, with the flaring causing a similar effect as the bifurcations, and the tapering taking the role of the terminations in our model. Given that synaptic boutons are several orders of magnitude smaller in NL and cortex, we do not expect a strong effect in these systems.

Membrane currents flowing in boutons were studied by *Geiger and Jonas (2000)*, who recorded from the terminals of hippocampal mossy fibers and examined calcium and potassium conductances. The potassium conductances broadened the incoming spikes in an activity-dependent manner. This spike broadening is hypothesised to be mediated by slow inactivation of the potassium channels and takes place on a timescale of >100 ms, and is thus not relevant to the present study. Spike broadening could be captured in our model by incorporating in *Equation 1* a $\sigma_{\text{spike}}$ that is variable in time.

The calcium currents reported by *Geiger and Jonas (2000)* were further quantified by *Alle et al. (2009)*. Calcium currents were temporally overlapping and much smaller in amplitude than sodium and potassium currents. We therefore neglected calcium currents in our model.

Modeling the myelinated compartments, we assumed that they are purely passive and strongly insulated from the extracellular space. However, myelinated compartments do in fact express active conductances, in particular in the paranodal and juxtaparanodal region (*Chiu and Ritchie, 1981*; *Waxman and Ritchie, 1985*). Including such a detailed distribution of ion channels in our model could lead to a different shape of the waveform and the spectrum of the EFP of an action potential, possibly similar to the effect described by *Ness et al. (2016)* for active conductances on dendrites. The conclusions drawn by our model are, however, independent of the precise active conductances and the distribution of myelinated and active segments along the axons because our results rely only on the gross waveform of propagating action potentials, but not on finer details. Active conductances and capacitive currents in the myelinated segments could affect the shape of the action potential waveform, but do not affect our conclusion about the spatial scaling behaviour of the EFP.

Because of the weak dependence of our results on the gross extracellular spike waveform, our analytical model does not include any intrinsic low-pass filtering as can be derived, for example, for dendritic models (*Lindén et al., 2010*; *Einevoll et al., 2013* ; for reviews see *Buzsáki et al., 2012*). The effective additional currents flowing at bifurcations and terminations are, however, low-frequency contributions to the overall membrane currents in our model. Extending our model to treat these currents separately might show whether axons could contribute to the observed $1/f$ scaling of the spectrum of the EFP (*Pritchard, 1992*).

### Conclusion

Axonal projections can contribute substantially to EFPs. Our results quantitatively show how the anatomy of axon terminal zones and the activity in axons determine its frequency-specific far-field contribution to the EFP.

## Materials and methods

### Experimental recordings

The experiments were conducted at the Department of Biology of the University of Maryland. Data was collected from three barn owls (*Tyto furcata pratincola*). Procedures conformed to NIH Guidelines for Animal Research and were approved by the Animal Care and Use Committee of the University of Maryland. Anaesthesia was induced prior to each experiment by intramuscular injection of a total of $8-10$ ml/kg of $20\%$ urethane divided into three to four injections over the course of $3$ hours. Body temperature was maintained at $39°$C by a feedback-controlled heating blanket.

Recordings were made in a sound-attenuating chamber (Acoustic Systems Inc., Austin, TX, USA). Tungsten electrodes with impedances between $2$ and $20$ MΩ were used to find suitable recording locations in nucleus laminaris (NL). Once NL had been located, the tungsten electrode was retracted and replaced with a 32 channel multi-electrode array (A1 $\times$ 32–15 mm-50-413-A32, Neuronexus, Ann Arbor, MI, USA). The multi-electrode array was lowered using a microdrive (MP225, Sutter Instruments Co., Novato, CA, USA) during continuous presentation of a white-noise burst stimulus until visual inspection of the waveform showed that NL was at the center of the array. A grounded silver chloride pellet, placed under the animal's skin around the incision, served as the animal ground electrode. Electrode signals were amplified by a headstage (HS36, Neuralynx Inc., Tucson, AZ, USA). An adapter (ADPT-HS36-N2T-32A, Neuralynx Inc.) was used between the electrode and the headstage. A further adapter (ADPT-HS-36-ERP-27, Neuralynx Inc.) was used between the headstage and the control panel in order to map all 32 channels to the amplifiers. Pre-amplified electrode signals were passed to the control panel (ERP27, Neuralynx Inc.), then to four 8-channel amplifiers (Lynx-8, Neuralynx Inc.), and then to an analogue-to-digital converter (Cheetah Digital Interface, Neuralynx Inc.) connected to a personal computer running Cheetah5 software (Neuralynx Inc.) where the responses were stored for off-line analysis.

Acoustic stimuli were digitally generated by a custom-made Matlab (MathWorks, Natick, MA, USA; RRID:SCR_001622) script (*McColgan and Liu, 2017*) driving a signal-processing board (RX6, Tucker-Davis Technologies, Alachua, FL, USA) at a sampling rate of 195.3125 kHz. The sound stimuli were attenuated using a programmable attenuator (PA5, Tucker-Davis Technologies). Click stimuli were generated as a single half-wave of a 5 kHz sine tone. Miniature earphones were inserted into the owl's ear canals and fixed to a headplate. Acoustic stimuli were fed to these earphones. Stimulus delivery was triggered by National Instruments equipment (NI USB-6259 and BNC-2090A, National Instruments Inc, Austin, TX, USA), and stimulus presentation times were recorded along with the responses. Trigger pulses were configured in MatLab through Ephus software (Vidrio Technologies LLC, Ashburn, VA, USA). Responses were averaged over 10 repetitions.

### Multi-compartment model of axons

We modeled axons using NEURON (*Hines and Carnevale, 1997*; *Hines et al., 2009*) and extended previous work by *Simon et al. (1999)* and *Kuba and Ohmori (2009)*, which included the high- and low-threshold potassium channels used by *Rathouz and Trussell (1998)*. The axon was modeled as a sequence of active nodes and passive myelinated segments. The nodes and myelinated segments had lengths of 2 μm and 75 μm, respectively. We used the model of a nucleus magnocellularis (NM)

axon provided by *Simon et al. (1999)* in ModelDB (*Hines et al. (2004)*, Accession number: 153998). In order to ensure robust spike propagation at the bifurcations, some of the model the parameters were modified. The values of the properties that were modified from those provided by *Simon et al. (1999)* are shown in *Table 1*. In addition, the Q10 values were set to 3, and the temperature was set to $40° \mathrm{C}$ as done by *Kuba and Ohmori (2009)*. The ratio of leak conductance and capacitance between node and myelin was changed from 80 as used by *Simon et al. (1999)* to 1000 (*Koch, 2004*). We removed the Hodgkin-Huxley type potassium conductivities included by *Simon et al. (1999)* (which are based on data from the squid axon) from the simulations, and increased the KLVA and KHVA conductances (which are based on physiological data from the auditory brainstem) to compensate. In order to avoid nodes lining up in axial direction, the very first myelinated segment had a length drawn from a uniform distribution between 0 and 75 µm.

We included branching axons in our simulations. Branches were generated by connecting the incoming passive segment to one end of a node, and the two outgoing passive segments to the other end of the node, and then continuing the alternation of active and passive segments in each resulting branch. In *Figure 1A–D*, where the positions of bifurcations or terminations of axons were fixed, the last node was placed in the axon as usual, and the last myelinated segment was shortened in order to obtain the total length before the bifurcation or termination.

To approximate the infinitely long axons, we added segments before and after the shown portions of the axon. We chose the total length of the additional segments by incrementally increasing the length segment-by-segment until there was no visual difference between each successive lengthening of the axon, arriving at a length of 3 mm at each end.

To evoke an action potential at a designated time, a special conductance was temporarily activated in the first node of Ranvier. The conductance had a reversal potential of 0 mV, a maximal amplitude of 0.05 µS, and a time course described by an alpha-function with time constant 0.01 ms. Soma and axon initial segment were not modeled explicitly. This conductance resembled a synaptic conductance, except for the very short time constant and the lack of a somatic or dendritic compartment on which synapses typically impinge.

For the simplified axon geometries used in *Figure 1*, the branching pattern was fixed as described in the caption, with the exception of the axial positions of the branching points in *Figure 1E*: a random offset between branching points was drawn from a gamma distribution with mean 400 µm and standard deviation 300 µm. The initial branching point for each axon was offset from the original location by a distance drawn from a Gaussian distribution with mean zero and a width of 300 µm. This was done to smooth out the effects of individual branchings or terminations.

For the axons in *Figure 2*, branching patterns were generated procedurally, starting with a root segment. In order to avoid artifacts from the stimulus and to simulate a long fiber tract prior to the terminal zone, a sequence of 10 active and passive segments without bifurcations was assumed before the terminal zone (770 µm total length). To this root, segments were appended iteratively. Before adding a segment, a decision whether to branch or terminate an axon was drawn from a

**Table 1.** Parameter values used for the multi-compartment model which were modified from those used by *Simon et al. (1999)*.

| Symbol | Meaning | Value | Value used by *Simon et al. (1999)* |
|---|---|---|---|
| $R_a$ | axial resistance | 50 $\Omega$ cm | 200 $\Omega$ cm |
| $\bar{g}_{\mathrm{Na}}$ | maximum sodium conductance | 2.4 S/cm2 | 0.32 S/cm$^2$ |
| $\bar{g}_{\mathrm{KLVA}}$ | maximum low-threshold potassium conductance | 0.1 S/cm2 | 3 mS/cm$^2$ |
| $\bar{g}_{\mathrm{KHVA}}$ | maximum high-threshold potassium conductance | 1.5 S/cm$^2$ | 30 mS/cm$^2$ |
| $g_{\mathrm{leak}}^{\mathrm{node}}$ | leak conductance in node | 1 mS/cm$^2$ | 0.28 mS/cm$^2$ |
| $g_{\mathrm{leak}}^{\mathrm{myelin}}$ | leak conductance in myelin | 1 S/cm$^2$ | 35 S/cm$^2$ |
| $E_{\mathrm{leak}}$ | leak reversal potential | −72 mV | −45 mV |
| $E_{\mathrm{K}}$ | potassium reversal potential | −80 mV | −60 mV |
| $E_{\mathrm{Na}}$ | sodium reversal potential | 50 mV | 40 mV |
| $c_m^{\mathrm{myelin}}$ | membrane capacitance in myelin | 1 nF/cm$^2$ | 12 nF/cm$^2$ |

DOI: https://doi.org/10.7554/eLife.26106.007

probability distribution that was dependent on the axial position of the end of the previous segment. These probability distributions were modeled as logistic functions with the parameters adjusted to roughly match the numbers of branchings and terminations shown by *Carr and Konishi (1990)*. Thus, an initial phase dominated by bifurcations was followed by a phase dominated by terminations, with the probability of termination reaching 100% at the end of the terminal zone. The distribution of bifurcations had its maximum at axial location $z = 0$ with a standard deviation of 200 μm. The distribution of terminations had its maximum at $z = 500$ μm, with a standard deviation of 100 μm. The branching angle had an average of $20°$, with a standard deviation of $5°$. At branching points, the plane containing the branches had a uniform random orientation, resulting in a 3-dimensional structure of the axon bundle.

In all cases except for the simulations shown in *Figure 2*, the radial position of all node and myelin compartments was set to zero, meaning they were placed on a straight line extending axially.

Numerical simulations of action potentials propagating along axons yielded the membrane currents from which we calculated extracellular fields. This procedure is described in detail by *Gold et al. (2006)*. Briefly, the extracellular medium is assumed to be a homogeneous volume conductor with conductivity $\sigma_e$, and a quasi-static approximation of the electrical field potential $\phi$ is made. The extracellular potential $\phi(\mathbf{r}, t)$ at location $\mathbf{r}$ and time $t$ due to a membrane current density distribution $i(\mathbf{r}, t)$ is then governed by the equation $\Delta\phi(\mathbf{r}, t) = \frac{1}{\sigma_e} i(\mathbf{r}, t)$, with $\Delta$ denoting the Laplace operator *Nunez and Srinivasan (2006)*. If the currents $i$ are constrained to a volume $W$, this equation has the solution

$$\phi(\mathbf{r}, t) = \frac{1}{4\pi\sigma_e} \int_W \frac{i(\mathbf{r}', t)}{|\mathbf{r} - \mathbf{r}'|} d\mathbf{r}' .$$  (2)

Since the majority of the current flows through the small nodes of Ranvier in myelinated axons, we used the point-source approximation for all compartments, and subdivided the myelinated segments into 10 iso-potential sections each; we did not use the line-source approximation (*Holt and Koch, 1999*).

Analysis of the resulting extracellular field potential (EFP) included filtering. All filtering was performed with third-order Butterworth filters. The multi unit activity (MUA) was calculated by high-pass filtering the signal with a cutoff of 2500 Hz, setting all samples with negative values to zero, and then low-pass filtering the resulting response with a cutoff of 500 Hz. The low-pass filtered EFP was calculated with a cutoff of 1000 Hz. To exclude influences of spectral leakage on our results, we also performed the same analysis with 20th-order Butterworth filters and found qualitatively identical results. The specific filtering (3rd or 20th order Butterworth) did not affect our conclusions.

The code for these simulations is available at https://github.com/phreeza/pyLaminaris (*McColgan and Liu, 2017*). A copy is archived at https://github.com/elifesciences-publications/pyLaminaris.

## Mean-field model of an axon bundle

To better understand the processes leading to the complex spatio-temporal patterns of extracellular fields, we devised an analytically tractable model of axon bundles. We defined the spatial dimension in cylindrical coordinates $\mathbf{r} = (\rho, \varphi, z)$, where $\rho$ was the radial distance from the cylindrical axis, $\varphi$ the angle of azimuth, and $z$ the axial distance along the cylinder axis. We considered a simple model axon bundle that extended infinitely in the axial $z$-direction at $\rho = 0$. The bundle had a variable number of fibers along the $z$ axis, denoted by $n(z)$, each of which cylindrical with an identical radius $a$. This meant that the total cross-sectional area $A$ of the bundle at a given depth $z$ was given by $A(z) = \pi a^2 n(z)$. We assumed the axons to be perfect transmission lines, meaning that the action potential is a traveling wave with velocity $v$ along the axon. In particular, we neglected delays and distortions that can be induced when an axon branches or terminates. In this case, we could assume that the membrane voltage was the same in each fiber for a given $z$ coordinate. From linear cable theory (e.g. *Jack et al., 1975*), we then obtained the following expression for the total transmembrane current per unit length $I(z, t)$ from a given membrane potential $V(z, t)$:

$$I(z,t) = \frac{\partial}{\partial z}\left(\frac{A(z)}{r_L}\frac{\partial}{\partial z}V(z,t)\right) \tag{3}$$

$$= \frac{\pi a^2}{r_L}\frac{\partial}{\partial z}\left(n(z)\frac{\partial}{\partial z}V(z,t)\right) \tag{4}$$

$$= \frac{\pi a^2}{r_L}\left(\frac{\partial}{\partial z}n(z)\cdot\frac{\partial}{\partial z}V(z,t) + n(z)\cdot\frac{\partial^2}{\partial z^2}V(z,t)\right) \tag{5}$$

We next calculated the corresponding extracellular field potential $\phi(\mathbf{r},t)$ of a given membrane potential waveform $V(z,t)$ propagating through the axon bundle. Due to the rotational symmetry of the system and the fact that current flows only at $\rho = 0$, the membrane current can be described as $i(\mathbf{r},t) = I(z,t)\frac{\delta(\rho)}{\rho}$, where $\frac{\delta(\rho)}{\rho}$ is the Dirac delta distribution for a line at $\rho = 0$. Applying this membrane current to *Equation 2*, we obtained

$$\phi(\mathbf{r},t) = \frac{1}{4\pi\sigma_e}\int_{-\infty}^{\infty}\frac{I(z',t)}{\sqrt{(z-z')^2 + \rho^2}}dz' \,, \tag{6}$$

which is independent of the angle $\varphi$.

To derive the frequency responses in *Figure 3*, we simulated the membrane potentials as sinusoids, that is, $V(z,t) = \sin(2\pi f \cdot (z - tv))$, with varying frequencies $f$ between 100 Hz and 5 kHz and calculated the standard deviation of the response for each frequency individually. The amplitude of the membrane potential $V$ was the same for all frequencies.

To derive the dipole moment of a simplified projection zone, we considered an axon bundle in which identical spikes with the waveform $V_{\text{spike}}(z,t)$ propagate as traveling waves with a velocity $v$ in positive $z$ direction: $V_{\text{spike}}(z,t) = V_{\text{spike}}(z - tv, 0)$. If each of the fibers is stimulated with an inhomogeneous Poisson process, with the driving rate $\lambda(t)$ shared among all axons, the average membrane potential across fibers will be $V(z,t) = V_{\text{spike}}(z,t)\star\lambda(t)$, where $\star$ denotes the convolution with respect to the time $t$. Plugging this into *Equation 5*, we obtained

$$I(z,t) = \frac{\pi a^2}{r_L}\left(\frac{\partial}{\partial z}n(z)\cdot\frac{\partial}{\partial z}\left[V_{\text{spike}}(z,t)\star\lambda(t)\right] + n(z)\cdot\frac{\partial^2}{\partial z^2}\left[V_{\text{spike}}(z,t)\star\lambda(t)\right]\right) \tag{7}$$

$$= \frac{\pi a^2}{r_L}\lambda(t)\star\left(\frac{\partial}{\partial z}n(z)\cdot\frac{\partial}{\partial z}V_{\text{spike}}(z,t) + n(z)\cdot\frac{\partial^2}{\partial z^2}V_{\text{spike}}(z,t)\right) \,. \tag{8}$$

## Analytical solution of the mean-field model of an axon bundle

Assuming Gaussian shapes for the firing-rate pulse $\lambda(t) = \bar{\lambda}_{\text{pulse}}\exp\left(-\frac{t^2}{2\sigma_{\text{pulse}}^2}\right)$, the spike $V_{\text{spike}}(z,t) = V_{\text{spike}}(z - tv, 0) = \bar{V}_{\text{spike}}\exp\left(-\frac{(z-tv)^2}{2\sigma_{\text{spike}}^2}\right)$, and the fiber bundle projection zone $n(z) = \bar{n}\exp\left(-\frac{z^2}{2\sigma_{\text{n}}^2}\right)$, we were able to take advantage of the fact that the product and the convolution of two Gaussians are again Gaussian, and obtained

$$I(z,t) = \bar{n}\bar{\lambda}_{\text{pulse}}\bar{V}_{\text{spike}}\sqrt{2}\pi^{3/2}a^2\cdot\exp\left(-\frac{z^2}{2\sigma_n^2} - \frac{(z-tv)^2}{2v^2\left(\sigma_{\text{pulse}}^2 + \sigma_{\text{spike}}^2\right)}\right) \tag{9}$$

$$\cdot\frac{\sigma_n^2\left(-v^2\sigma_{\text{pulse}}^2 - v^2\sigma_{\text{spike}}^2 + (z-tv)^2\right) - v^2z\left(\sigma_{\text{pulse}}^2 + \sigma_{\text{spike}}^2\right)(tv-z)}{v^4 r_L\sigma_n^2\sqrt{\frac{1}{\sigma_{\text{pulse}}^2} + \frac{1}{\sigma_{\text{spike}}^2}}\left(\sigma_{\text{pulse}}^2 + \sigma_{\text{spike}}^2\right)^2} \,.$$

The dipole moment $p(t)$ is defined as

$$p(t) = \int_{-\infty}^{\infty}z\cdot I(z,t)dz \,, \tag{10}$$

into which we can enter *Equation 9* to obtain

$$p(t) = -\bar{n}\bar{\lambda}_{\text{pulse}}\bar{V}_{\text{spike}}\frac{2\pi^2 a^2}{r_L}\frac{v^2\sigma_n\sigma_{\text{pulse}}\sigma_{\text{spike}}}{\left(\sigma_n^2 + v^2\left(\sigma_{\text{pulse}}^2 + \sigma_{\text{spike}}^2\right)\right)^{3/2}} \cdot t\exp\left(\frac{-t^2 v^2}{2\left(\sigma_n^2 + v^2\left(\sigma_{\text{pulse}}^2 + \sigma_{\text{spike}}^2\right)\right)}\right) \quad . \quad (11)$$

The dipole moment has its peak amplitude at $t_{\max} = -\sqrt{\sigma_n^2 + v^2\left(\sigma_{\text{pulse}}^2 + \sigma_{\text{spike}}^2\right)}/v$, and takes the value

$$p_{\max} = p(t_{\max}) = \frac{2a^2\pi^2}{r_L\sqrt{e}}\cdot\frac{v\sigma_n\sigma_{\text{pulse}}\sigma_{\text{spike}}\bar{n}\bar{\lambda}_{\text{pulse}}\bar{V}_{\text{spike}}}{\left(\sigma_n^2 + v^2\left(\sigma_{\text{pulse}}^2 + \sigma_{\text{spike}}^2\right)\right)} \quad , \quad (12)$$

which is presented as *Equation 1* in the Results section.

## Model fitting to experimental data

In order to relate the model to experimentally obtained data as shown in *Figure 5*, we performed a nonlinear least squares fit, minimizing the mean squared error $\epsilon$ between the measured potential $\phi_{\text{measured}}$ and the model prediction $\phi_{\text{model}}$ in *Equations 6 and 5* for $N = 32$ measurement locations $z_n$ ($n = 1, \ldots, N$) and $M = 600$ time points $t_m$ ($m = 1, \ldots, M$): $\epsilon = \frac{1}{NM}\sum_{n=1}^{N}\sum_{m=1}^{M}[\phi_{\text{measured}}(z_n, t_m) - \phi_{\text{model}}(z_n, t_m)]^2$. The separation between electrodes was given by the electrode layout as 50 µm. The time between sampling points was 5.12 µs. We achieved the minimization of the error $\epsilon$ using the 'optimize.minimize' routine provided by the SciPy package (*Jones et al., 2001*). The free parameters to be determined by the optimization routine were the distance $\rho$, the velocity $v$, the number of fibers per unit length $n(z_n)$ for each measurement location $z_n$, and the spatial derivative of the average membrane potential $\frac{\text{d}}{\text{d}z}V(z_1, t_m)$ at electrode location $z_1$ for each time point $t_m$. We fit the first derivative of the membrane potential in order to better capture the low-frequency components that we found in *Figure 1E*, and because the membrane potential appears only as the derivative in the model. The derivative of the membrane voltage at the other locations than $z_1$ was then determined by the traveling-wave assumption: $\frac{\text{d}}{\text{d}z}V(z_n, t_m) = \frac{\text{d}}{\text{d}z}V(z_1, t_m - \frac{z_n - z_1}{v})$, using a linear interpolation between timepoints. The model assumption of a single line of axons with electrodes at a fixed distance is a simplification of a three-dimensional axon tree where the fibers are distributed at various distances. The distance parameter $\rho$ in *Equation 6* can be interpreted as an average distance in this simplification.

To aid the convergence of the fit algorithm, an initial guess for the number of fibers $n(z_n)$ was set by hand. Initializing the guess to a constant or a fully random number of fibers resulted in a failure to converge. However, different Gaussian-like initial guesses converged to a single solution, meaning that the specific initial guess did not alter the final fit result. Initializing the membrane voltage with different normally distributed values did not affect the outcome of the fit. The results shown in *Figure 5* were obtained with an initial guess of a Gaussian with amplitude 12, centered at penetration depth 725 µm with standard deviation 400 µm.

Because of the linearity of *Equations 2-6* both in the current $I$ and the membrane potential $V$, inferring the membrane voltage $V$ from the average over trials of the extracellular potential $\phi$ produces the average membrane voltage $V$. This in turn is the membrane voltage response of a single spike convolved with the peri-stimulus time histogram (PSTH).

## Acknowledgements

This work was supported by the German Federal Ministry for Education and Research Grants 01GQ1001A and 01GQ0972, NIH DCD 000436 and US-American Collaboration in Computational Neuroscience 'Field Potentials in the Auditory System' as part of the NSF/NIH/ANR/BMBF/BSF Collaborative Research in Computational Neuroscience Program, 01GQ1505A. The authors wish to thank Anna Kraemer for help with animal handling and surgery; and Martina Michalikova and Tiziano D'Albis for helpful comments on a draft of this manuscript.

## Additional information

### Competing interests
Catherine Emily Carr: Reviewing editor, *eLife*. The other authors declare that no competing interests exist.

### Funding

| Funder | Grant reference number | Author |
| --- | --- | --- |
| National Institute on Deafness and Other Communication Disorders | DC-00436 | Catherine Emily Carr |
| German Ministry for Education and Research | 01GQ1001A | Richard Kempter |
| Collaborative Research in Computational Neuroscience Program | 01GQ1505A | Richard Kempter |
| National Science Foundation | 1516357 | Catherine Emily Carr |

The funders had no role in study design, data collection and interpretation, or the decision to submit the work for publication.

### Author contributions
Thomas McColgan, Conceptualization, Software, Formal analysis, Investigation, Visualization, Methodology, Writing—original draft, Writing—review and editing; Ji Liu, Resources, Software, Investigation, Methodology, Writing—review and editing; Paula Tuulia Kuokkanen, Hermann Wagner, Conceptualization, Investigation, Writing—review and editing; Catherine Emily Carr, Conceptualization, Resources, Supervision, Funding acquisition, Investigation, Methodology, Writing—review and editing; Richard Kempter, Conceptualization, Supervision, Funding acquisition, Investigation, Project administration, Writing—review and editing

### Author ORCIDs
Thomas McColgan  http://orcid.org/0000-0003-0027-0580
Paula Tuulia Kuokkanen  http://orcid.org/0000-0003-1355-227X
Catherine Emily Carr  http://orcid.org/0000-0001-5698-6014
Hermann Wagner  http://orcid.org/0000-0002-8191-7595
Richard Kempter  http://orcid.org/0000-0002-5344-2983

### Ethics
Animal experimentation: This study was performed in accordance with the recommendations in the Guide for the Care and Use of Laboratory Animals of the National Institutes of Health. All of the animals were handled according to approved institutional animal care and use committee (IACUC) protocols R-13-14 and R-16-11 of the University of Maryland. All experiments were non-recovery, with surgery performed under urethane anesthesia."

### Decision letter and Author response
Decision letter https://doi.org/10.7554/eLife.26106.009
Author response https://doi.org/10.7554/eLife.26106.010

## Additional files

### Supplementary files
• Transparent reporting form
DOI: https://doi.org/10.7554/eLife.26106.008

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
