## [Decision Letter]

Thank you for submitting your article "Dipolar extracellular potentials generated by axonal projections" for consideration by *eLife*. Your article has been reviewed by Timo van Kerkoerle (Reviewer #1); Gaute T Einevoll (Reviewer #2); and Rishikesh Narayanan (Reviewer #3), and the evaluation has been overseen by a Reviewing Editor and Andrew King as the Senior Editor.

The reviewers have discussed the reviews with one another and the Reviewing Editor has drafted this decision to help you prepare a revised submission.

Summary:

The manuscript investigates the contribution of action potentials to the local field potential, and shows how, contrary to common wisdom, axonal arbors can generate dipolar fields. The manuscript is well written and the figures are generally clear.

Essential revisions:

The revisions requested are of two-fold nature: 1) more exploration of the generality of the effects, 2) more justification and exploration of the findings.

As mentioned in the manuscript an important factor in the contribution of action potentials is the synchronicity between neighboring axons; if there is too much jitter in the spikes coming in on neighboring axons the fields generated by them will cancel it. It would be of interest to quantify what the synchronicity needs to be on neighboring axons to be able to see a significant effect in the LFP. Alternatively, for a fixed level of coherence between the spikes, what would the generated field be for different frequencies? It can be judged from Figure 3 that even for bifurcations on the same axon, the frequency has to be high to still see an effect. It would therefore be relevant to see for a realistic level of coherence between the spikes what the generated axonal field would be for different frequencies. This can help to understand whether this phenomenon could play a role in (1) high-frequency ripples, as for example measured with ECoG (Worrell et al. Brain 2008), or even in (2) high-frequency gamma activity, as measured as dipoles within the cortex (van Kerkoerle et al. PNAS 2014), or with MEG on the scalp (e.g. Siegel et al. Neuron 2008).

This result will also make it easier to judge how to relate it to the previous literature (which part of the LFP can be attributed to dipoles generated by axonal arbors), and thereby how relevant the current manuscript is for a broader audience.

The auditory brainstem of the barn owl is a rather special case where the activation is very brief, in the order of a few milliseconds. The authors also refer to early papers in the visual cortex (Mitzdorf and Singer 1978; Mitzdorf, 1985), but these are special cases where electrical stimulation of the optic radiation was used, which also gives a dipolar field in the order of milliseconds as well. In contrast, the dipolar fields as evoked by a visual stimulus, are generally in the order of tens of milliseconds (e.g. Mitzdorf, 1985; Schroeder et al., 1991; Schroeder, Mehta and Givre, 1998; Self et al., 2013). It would be relevant to understand what part of a more common visual activation like that can be attributed to axonal fields.

The paper makes it conceivable that population axonal spikes contribute to dipoles which would be measurable in the LFP and potentially even to scalp EEG/MEG.

Can the authors extract the dipole and quadrupole moments from the fit?

Can the authors assess the uniqueness of the fit in their optimization procedure? Could there be several disparate solutions to this fitting procedure, possibly resulting in significant variability in the computed moments based on the specific fit that was employed?

An improved comparison is needed between the numerically comprehensive scheme and the simplified analytical model. It would be nice to illustrate how similar they are for a well-chosen example.

"All filtering was performed with third-order Butterworth filters […] The LFP was calculated by low-pass filtering the signal with a cut-off of 1000 Hz."

The filter used by the authors to obtain the LFP is a 3rd order Butterworth with a cut-off of 1000 Hz. The cut-off frequency is large, in comparison to most studies employing a value set at 100 Hz to 500 Hz (Buzsaki, Anastassiou and Koch, 2012; Schomberg et al., 2012; Taxidis et al., 2015). Additionally, as the order of the filter is 3, the sharpness of the filter transition from the pass band to the stop band would be shallow. This implies that frequencies beyond 1000 Hz also pass through. Additionally, in some figures, the low frequency component is arbitrarily represented at an even higher 2 kHz. EFP and LFP are alternately employed, and there is a lot of ambiguity with reference to notation and terminology. The specific choice for cutoff frequencies is of utmost importance because the authors' central claim about the contributions of (high frequency) action potentials propagating through the axon to LFPs. How dependent is the filter structure for the conclusions drawn by the authors? Would the conclusions be different if the authors employed, say a 20th order Butterworth filter with a 250 Hz cutoff?

In describing the model fit, it would be valuable to describe the most important assumptions made. Also other details of modeling need attention:

a) Location of nodes of Ranvier: In the case where there were multiple axons, were the nodes of Ranvier placed at the same spatial axis?

b) "Note that in order to avoid confounding effects, the horizontal distances between axons in Figure 1 are for illustration only; all collaterals were simulated to lie on a straight line." Isn't this an oversimplification that artificially boosts contributions of the transmembrane currents from different axons to the EFP? What happens if this is changed to reflect an anatomical organization where horizontal distances do exist?

c) What was the structure of the axon at the bifurcations – was the surface area larger at bifurcations? What was the location of the terminal with reference to the nearest node? What would be the impact of terminating axons with a bouton with appropriate dimensions containing all ion channels (calcium, sodium and potassium especially) they are supposed to express? How would the bouton geometry (Gydikov et al., 1984 and Gydikob and Trayanova, 1986) and ion channel densities there, along with changes in each parameter mentioned here, alter the conclusions?

d) The authors assume that the myelin sheath does not contribute to the EFPs. However, given the several channels and receptors that these glial cells express, and given the active roles they play during action potential propagation, is it reasonable to assume that they don't contribute to the EFPs? What would be the contribution of the capacitative current that is consequent to the dielectric myelin between two parallel conducting media, with a high-frequency action potential propagating across? Is there any contribution of the 1/f^n^ scaling that is generally observed in LFPs, especially given the absence of intrinsic dendritic filtering in the analysis here (Nature Reviews Neuroscience reviews of Buzsaki et al., 2012, Einevoll et al., 2013)? Could the capacitive current (with reference to the myelinated compartments), however small, contribute to such filtering?

e) Please provide details of how the infinite axon was simulated.

---

## [Author Response]

As mentioned in the manuscript an important factor in the contribution of action potentials is the synchronicity between neighboring axons; if there is too much jitter in the spikes coming in on neighboring axons the fields generated by them will cancel it. It would be of interest to quantify what the synchronicity needs to be on neighboring axons to be able to see a significant effect in the LFP. Alternatively, for a fixed level of coherence between the spikes, what would the generated field be for different frequencies? It can be judged from Figure 3 that even for bifurcations on the same axon, the frequency has to be high to still see an effect. It would therefore be relevant to see for a realistic level of coherence between the spikes what the generated axonal field would be for different frequencies.

Synchronicity between the action potentials in neighboring axons is indeed a key factor in our manuscript. One way to quantify synchronicity is the width of a population pulse. This approach was used throughout the text and in all Figures (except Figure 3). One of our main messages of the manuscript is that the spatial width of this population pulse should match the length of the terminal zone in order to generate the maximum dipolar contribution to the EFP. The explicit dependence of the dipole moment on the synchronicity (of a pulse) is given in Equation 1, where the symbol σ_pulse_ describes the width of the pulse.

Another way to study synchronicity is axonal activity oscillating at a particular frequency, as used in Figure 3. Such an oscillating axonal activity that is travelling down an axonal bundle at a particular velocity leads to a spatial oscillation pattern along the axon. This pattern can generate a dipole moment that depends on the spatial wavelength. The dipolar component in the EFP achieves a maximum if the spatial wavelength matches the width of the terminal zone (Figure 3); and the dipole moment approaches zero in the limits of very small and much larger frequencies.

For a fixed spatial wavelength, one could also modify the coherence of the oscillation, which is also called the "strength of phase locking" or the "vector strength". Even though the generated dipole naturally increases with increasing vector strength, the shapes of the curves in all graphs in Figure 3 are independent of phase locking because they were normalized or show ratios. In this way we intended to disentangle the impact of the frequency on the dipole moment from the effect of coherence.

In order "to see for a realistic level of coherence between the spikes what the generated axonal field would be for different frequencies", we would like to refer to Equation 1, which quantifies the impact of the width of pulse packets on the dipole moment.

Moreover, to better illustrate the relation of coherence and frequency in Figure 3, we added in panels A and B sketches that show the spatial wavelength of the oscillation with respect to the terminal zone. In Figure 3 we added (to the horizontal frequency axis) an axis that describes the wavelength.

This can help to understand whether this phenomenon could play a role in (1) high-frequency ripples, as for example measured with ECoG (Worrell et al. Brain 2008), or even in (2) high-frequency gamma activity, as measured as dipoles within the cortex (van Kerkoerle et al. PNAS 2014), or with MEG on the scalp (e.g. Siegel et al. Neuron 2008).This result will also make it easier to judge how to relate it to the previous literature (which part of the LFP can be attributed to dipoles generated by axonal arbors), and thereby how relevant the current manuscript is for a broader audience.

We think that it is unlikely that locally and intrinsically generated oscillations such as hippocampal ripples or γ will contribute to local EFPs, EcOGs, or EEGs of axonal origin because the resulting spatial wavelengths are rather long. For example, for 200 Hz ripple oscillations and ~0.5 m/s conduction velocity (Schaffer collaterals) the spatial wavelength is ~2.5 mm, which is much larger than typical projection zones in the hippocampal formation. Action potentials contribute more to the EFP for higher frequencies (or shorter pulses), which are more relevant for sensory driven events, for example in the auditory pathway (measured as the auditory brainstem response, ABR).

To address this issue, we have added in the Discussion at the end of the section “Dipolar EFPs in other animals and brain regions” a sentence that summarizes the dependence of pulse width, propagation velocity, and size of the projection zone, and there is also a related new paragraph at the end of the Results. Furthermore, the improved Figure 3 could aid in understanding.

The auditory brainstem of the barn owl is a rather special case where the activation is very brief, in the order of a few milliseconds. The authors also refer to early papers in the visual cortex (Mitzdorf & Singer 1978; Mitzdorf 1985), but these are special cases where electrical stimulation of the optic radiation was used, which also gives a dipolar field in the order of milliseconds as well. In contrast, the dipolar fields as evoked by a visual stimulus, are generally in the order of tens of milliseconds (e.g. Mitzdorf, 1985; Schroeder et al., 1991; Schroeder, Mehta and Givre, 1998; Self et al., 2013). It would be relevant to understand what part of a more common visual activation like that can be attributed to axonal fields.

ABRs and auditory evoked potentials (AEPs) are brief, but widely relevant for clinical and basic research on the auditory system. We hope our findings will be directly relevant to auditory studies. For longer pulses, with durations of 10 ms or more, the expected contributions to the EFP are indeed quite small. However, the dipole moment increases with decreasing axonal conduction velocity.

To better explain what the expected contributions to the EFP of more common visual stimulations are, we added a paragraph in the Results (new paragraph before the Discussion). We are also referring back to this point in the Discussion where we now outline more explicitly the impact of axonal contributions in other systems in the section "Dipolar EFPs in other animals and brain regions".

The paper makes it conceivable that population axonal spikes contribute to dipoles which would be measurable in the LFP and potentially even to scalp EEG/MEG.Can the authors extract the dipole and quadrupole moments from the fit?

We calculated the dipole moment from the data, as suggested, and added the value of ~3 µA mm we obtained to the manuscript in the last section of the Results.

We limited ourselves to the dipole moments. Calculating quadrupole moments directly from the currents would require an additional extensive discussion of the underlying theory. We think that such a discussion would add little information because the field potential at long distances will always be dominated by the dipolar component.

Can the authors assess the uniqueness of the fit in their optimization procedure? Could there be several disparate solutions to this fitting procedure, possibly resulting in significant variability in the computed moments based on the specific fit that was employed?

We attempted to assess the uniqueness of the fit by choosing different initial conditions to the fitting procedure. We added an explanation of this to the Materials and methods section “Model fitting to experimental data”. The combination of (1) the convergence of the fit from different starting points to similar fit results and (2) the fact that the fit results agree with known anatomy and activation statistics of the system give us the confidence that this is indeed the global optimum.

An improved comparison is needed between the numerically comprehensive scheme and the simplified analytical model. It would be nice to illustrate how similar they are for a well-chosen example.

We compared EFPs generated by (A) the numerical model as shown in Figure 2, and (B) the analytical model with a matched activation and fiber distribution. In Author response image 1 we show the heatmap of the difference ratio |A-B|/(|A|+|B|), averaged over time. The fiber bundle lies along the horizontal axis and terminates at axial distance 16500 µm. Spikes propagate from left to right.

The relative difference at distances greater than 1 mm was < 0.3 in radial direction and < 0.05 in axial direction. The small bump that radiates from the projection zone in radial direction (at axial distance ~15500 µm) is a spurious dipole. This spurious dipole is due to a differential activation of fibers that are radially separated.

We have mentioned this consistency check in the Results in the subsection "Effects of bifurcations and terminations on distance scaling of EFPs".

The spurious dipole is also further discussed in our reply to the question about under- or overestimating contributions by removing the radial dimension in the axon layout.

"All filtering was performed with third-order Butterworth filters […] The LFP was calculated by low-pass filtering the signal with a cut-off of 1000 Hz."The filter used by the authors to obtain the LFP is a 3rd order Butterworth with a cut-off of 1000 Hz. The cut-off frequency is large, in comparison to most studies employing a value set at 100 Hz to 500 Hz (Buzsaki, Anastassiou and Koch, 2012; Schomberg et al., 2012; Taxidis et al., 2015). Additionally, as the order of the filter is 3, the sharpness of the filter transition from the pass band to the stop band would be shallow. This implies that frequencies beyond 1000 Hz also pass through. Additionally, in some figures, the low frequency component is arbitrarily represented at an even higher 2 kHz. EFP and LFP are alternately employed, and there is a lot of ambiguity with reference to notation and terminology. The specific choice for cutoff frequencies is of utmost importance because the authors' central claim about the contributions of (high frequency) action potentials propagating through the axon to LFPs. How dependent is the filter structure for the conclusions drawn by the authors? Would the conclusions be different if the authors employed, say a 20th order Butterworth filter with a 250 Hz cutoff?

We thank the reviewer for pointing out the ambiguity of our usage of the term LFP and EFP. In the revised manuscript, we now refer to the low-frequency component as the "low-pass filtered EFP" instead of “LFP”, but point out that it is calculated in a way somewhat similar to the LFP, except for our usage of a relatively high cutoff frequency.

The reason that we chose relatively high cutoff frequencies in the kHz range was that our example simulations are meant to resemble the auditory brainstem, which is also where we obtained our data from. The optimal cutoff frequency depends on the system being studied. The auditory brainstem operates on short time scales, and the spectral frequencies are correspondingly high. In simulations with slower pulse and spike dynamics, the time scales would be slower, and frequencies would be lower.

We thus provide a general framework that can be applied to many systems and time scales, and an estimate for contributions in different scenarios given in Equation 1. Here one can see how spatial and temporal widths of the pulse, spike, and projection zone need to be modified to obtain dipolar contributions at any timescale. Furthermore, we added a paragraph to the subsection "Relevance to the interpretation of electrophysiological recordings" of the Discussion to clarify our chosen cutoff frequency.

To address the question of spectral leakage due to low filter order, we repeated the simulations for Figure 2 with a 20th order Butterworth filter (see Author response image 2), and found a very similar response to that from the 3rd order filter. This leads us to conclude that spectral leakage is not affecting the findings. In the revised manuscript, we added a note on the filtering in Materials and methods at the end of the "Multi-compartment model of axons".

**Author response image 2. respfig2:** 

Finally, we now filter in the same frequency bands throughout the manuscript (low frequency < 1 kHz, high frequency > 2.5 kHz). The detailed filter structure was not important for our conclusions.

In describing the model fit, it would be valuable to describe the most important assumptions made.

In the fourth paragraph in the Results subsection (about the barn owl data), we added a description of the model assumptions when describing the fit.

Also other details of modeling need attention:a) Location of nodes of Ranvier: In the case where there were multiple axons, were the nodes of Ranvier placed at the same spatial axis?

In the axial direction, the nodes were placed at randomized locations in all cases. They did not line up between multiple axons. This was achieved by randomizing the length of the first myelinated segment. We clarified this in the Materials and methods section.

In the simulations where the axons were placed along a line with no radial spread (all except Figure 2), the nodes were consequently also all on this line. This follows from the description in Materials and methods (section "Multi-compartment model of axons"), which was improved to clarify this point.

b) "Note that in order to avoid confounding effects, the horizontal distances between axons in Figure 1 are for illustration only; all collaterals were simulated to lie on a straight line." Isn't this an oversimplification that artificially boosts contributions of the transmembrane currents from different axons to the EFP? What happens if this is changed to reflect an anatomical organization where horizontal distances do exist?

Putting axons on a straight line has the effect of slightly *underestimating* the axonal contributions at all except very small (relative to the radial spread) distances from this line. There are two reasons for this:

First, a recording electrode at 100 µm from the trunk could randomly come arbitrarily close to a single fiber, leading to a very high contribution. This is not possible in our simplification. Second, if the currents in two fibers at the same depth but different radial offsets have a current with opposite polarity due to the random nature of spike generation, this would lead to an instantaneous nonzero dipole moment in the radial direction. This spurious dipole moment is different from the deterministic dipole moment discussed in the manuscript, but would lead to a stronger fluctuating contribution from the axons at increasing radial distances when compared to our simplified model (see Author response image 1). We thus consider our simplification to be a conservative one, helping us understand the axial dipole that we are interested in more clearly.

c) What was the structure of the axon at the bifurcations – was the surface area larger at bifurcations? What was the location of the terminal with reference to the nearest node?

We have clarified the corresponding section “Multi-compartment model of axons” in Materials and methods. Briefly, the surface area of the bifurcation was not larger than that of the remaining axon, but was formed by a normal node segment. To avoid propagation failures at branching points, we increased the ion channel densities throughout the axon compared to those used by Simon, Carr and Shamma, 1999 in a non-bifurcating axon (see new Table 1). The nearest node to the termination was determined by the normal succession of myelinated and node segments in the axon, with the last passive segment being truncated to obtain the desired overall length of the axon.

What would be the impact of terminating axons with a bouton with appropriate dimensions containing all ion channels (calcium, sodium and potassium especially) they are supposed to express? How would the bouton geometry (Gydikov et al., 1986 and Gydikov and Trayanova, 1986) and ion channel densities there, along with changes in each parameter mentioned here, alter the conclusions?

We added a new subsection titled "Relationships to more detailed biophysical models" to the Discussion addressing this point. We argued that including all ion channels in small boutons would not alter the conclusions.

d) The authors assume that the myelin sheath does not contribute to the EFPs. However, given the several channels and receptors that these glial cells express, and given the active roles they play during action potential propagation, is it reasonable to assume that they don't contribute to the EFPs? What would be the contribution of the capacitative current that is consequent to the dielectric myelin between two parallel conducting media, with a high-frequency action potential propagating across? Is there any contribution of the 1/f^n^ scaling that is generally observed in LFPs, especially given the absence of intrinsic dendritic filtering in the analysis here (Nature Reviews Neuroscience reviews of Buzsaki et al., 2012, Einevoll et al., 2013)? Could the capacitive current (with reference to the myelinated compartments), however small, contribute to such filtering?

We addressed this point in the new subsection "Relationships to more detailed biophysical models" of the Discussion. We agree that the myelin sheath could contribute to the EFP but we think that this does not compromise our main findings. In contrast, a 1/f^n^ scaling might boost far field dipolar axonal EFPs.

e) Please provide details of how the infinite axon was simulated.

We added an explanation to the first paragraph to the Materials and methods subsection "Multi-compartment model of axons" and clarified the terminology at the first usage of this term.